# Crane: Context-Guided Prompt Learning and Attention Refinement for Zero-Shot Anomaly Detection

## Abstract

Zero-shot anomaly detection/localization trains on a source domain and discriminates images from unseen target domains given only textual prompts (e.g., "normal" vs. "anomaly"); therefore, performance hinges on generalization. Recent methods build on CLIP for its strong zero-shot generalization; however, as we show, localization has not improved as much as detection and, especially for small regions, remains near random, with AUPRO close to chance, indicating weak pixel-level generalization. We attribute this to CLIP's limited ability to retain fine-grained features in its vision encoder and insufficient alignment between the text encoder and dense visual features, which have not been effectively addressed in previous methods. To address these challenges, first, we replace CLIP's vision encoder with an adapted vision encoder that uses a correlation-based attention module to better preserve fine-grained features and small details. Second, we boost text–vision alignment by conditioning the learnable prompts in the text encoder on image context extracted from the vision encoder and performing local-to-global representation fusion, further improving localization. Finally, we show that our correlation-based attention module can incorporate feature correlations from additional models such as DINOv2, further enhancing spatial understanding and localization. We call our model **Crane** (Context-Guided Prompt Learning and Attention Refinement) and its DINOv2-boosted variant $\texttt{Crane}^+$ and show that it improves the state-of-the-art by up to 28% in pixel-level localization (AUPRO) and up to 4.5% in image-level detection (AP), across 14 industrial and medical datasets.

## 1   Introduction

Image anomaly detection is the task where a set of training images representing normal visual patterns is given, and the goal is to identify test images that deviate from this notion of normality Ruff et al. (2021). It is particularly important in practical scenarios where abnormal samples are rare, diverse, and difficult to collect during training. For example, in medical diagnostics, datasets contain many healthy patient scans but few from those with rare conditions Salehi et al. (2021a). Similarly, in industrial defect detection Liu et al. (2024) and self-driving cars, normal data is abundant, but anomalies such as manufacturing defects or road obstacles are rare yet critical to detect Bogdoll et al. (2022). In these cases, a model is trained on normal data to identify abnormalities.

Recent studies Zhou et al. (2024); Cao et al. (2024); Deng et al. (2023); Jeong et al. (2023); Qu et al. (2024) show that collecting normal data from all target domains is often impractical, due to factors such as privacy concerns. To address this, zero-shot anomaly detection is introduced, allowing models trained on source data to detect anomalies in unseen target datasets without requiring domain-specific samples. Current methods either leverage CLIP's zero-shot capabilities with manually crafted prompts or adapt CLIP by learning prompts from source data to enable domain generalization. Despite progress, generalization remains inconsistent between image-level and pixel-level performance. For instance, as shown by Figure 1, two state-of-the-art methods, AdaCLIP Cao et al. (2024) and AACLIP Ma et al. (2025), get high image-level results while significantly compromising pixel-level performance (50.1% and 42.0% AUPRO) on industrial benchmarks, highlighting challenges in anomaly localization.

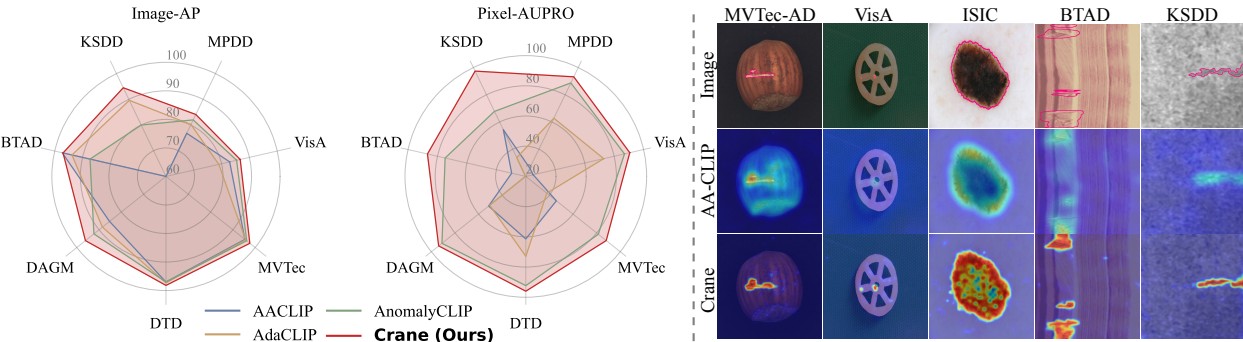

Figure 1: **Crane 's zero-shot anomaly detection performance compared to state-of-the-art methods.** The radar plots summarize image and pixel level performances, showing Crane consistently improves performance across industrial benchmarks. Qualitative examples further demonstrate that Crane produces more precise localizations than competing approaches. Anomalous regions outlined in pink. For complete qualitative comparison please refer to Appendix 6.

We attribute this gap to two key challenges: (1) Abnormal regions are often small, and the coarse-grained pretraining of CLIP limits the sensitivity of its global features to anomalous patterns, reducing its ability to distinguish normal samples from abnormal variations. (2) CLIP's dense features are not well-suited for segmentation tasks due to spatial misalignment Salehi et al. (2023); Wysoczańska et al. (2024); Salehi et al. (2025), limiting the model's ability to capture fine-grained details. Previous methods Zhou et al. (2024) address these challenges by prompt learning or fine-tuning the vision encoder on the source data. However, in prompt learning, the text encoder often fails to generate representations that are discriminative enough for subtle anomalies due to its limited capacity and the coarse pretraining of the text encoder. Fine-tuning the vision encoder, as done in prior work Cao et al. (2024); Lan et al. (2024a), can alleviate this issue in some domains, yet it can also limit generalization and lead to subpar performance in others, since the available training data is often limited and can encourage in-domain overfitting, ultimately hurting domain generalization.

To address these challenges more effectively, we propose Crane (Context-guided Prompt Learning and Attention Refinement). To learn more discriminative features for the text encoder, we guide prompts using the image classification token (CLS) from the image encoder, alongside other learnable parameters. This enables the model to generate representations conditioned on the image context, improving the modeling of fine-grained distributions in a data-efficient manner. Additionally, we introduce a local-to-global fusion mechanism that aggregates dense anomalous features into the global visual embedding, enhancing its sensitivity to local anomalous cues. To retain spatial alignment, we modify the CLIP vision encoder by introducing a correlation-based attention module that better captures fine-grained local information. Moreover, we introduce a simple method to inject spatial knowledge from vision encoders such as DINOv2 Oquab et al. (2023) into our zero-shot framework–even though these encoders are not inherently zero-shot–further improving performance. We call this variant Crane$^+$ . Our extensive evaluations across 14 datasets spanning medical and industrial domains show that Crane achieves strong performance relative to state-of-the-art approaches, with the most consistent gains observed on industrial benchmarks, including improvements of 2.3%–5.8% in anomaly detection and 2.6%–28% in localization, while Crane$^+$, delivers additional pixel-level gains of 0.3%–5.4%, further supporting its generalization ability for complex and fine-grained domains.

## 2 Related Works

**Unsupervised & Semi-supervised Anomaly Detection** Unsupervised and semi-supervised anomaly detection are dominantly used methods in the field; their main assumption is there is access to enough normal samples from the target domains. For unsupervised anomaly detection, a common strategy leverages pretrained models Salehi et al. (2021b); Liu et al. (2024) to extract discriminative features, modeling the normal distribution through mechanisms like knowledge distillation Salehi et al. (2021c); Deng & Li (2022);

Batzner et al. (2024), memory banks Roth et al. (2022); Gu et al. (2023), reconstruction-based methods Fang et al. (2023), and flow-based techniques Yu et al. (2021); Gudovskiy et al. (2022). As the anomaly data is unavailable during training, some methods generate synthetic anomalies using self-supervised methods Liu et al. (2023); Sträter et al. (2024), data augmentation Zou et al. (2022), or generative models Mirzaei et al. (2023); Hu et al. (2024); Chen et al. (2024a). Some works directly use the diffusion models to better model normal data, resulting in better detection Yao et al. (2024); Fučka et al. (2024). Semi-supervised anomaly detection methods incorporate a few anomalous samples during training Ruff et al. (2019); Pang et al. (2019); Yao et al. (2023); Zhang et al. (2023); Ding et al. (2022) to cope with the lack of abnormal samples during training. Although effective, these methods assume the availability of enough normal samples from the target domain, differing from our objective. In contrast, we evaluate generalization performance on a target dataset while exclusively training on source data independent of that target, which is explained in the zero-shot anomaly detection.

**Zero-shot anomaly detection** Zero-shot anomaly detection methods assume no access to the target dataset; instead, they leverage foundation models Radford et al. (2021); Li et al. (2022); Kirillov et al. (2023), pretrained on large-scale datasets, to learn generalizable features from source data that can be applied to unseen target datasets. In particular, contrastive vision-language models, e.g., CLIP Radford et al. (2021) which aligns global visual embeddings with textual descriptions. However, CLIP struggles with patch-level misalignment and lacks domain-specific sensitivity, limiting its ability to detect fine-grained anomalies. To address this issue, early methods focused on designing manually curated prompt templates Jeong et al. (2023); Deng et al. (2023); Chen et al. (2023); Cao et al. (2023), which depend on domain knowledge and prompt quality. Whereas more recent works adopted prompt learning techniques Zhou et al. (2022); Derakhshani et al. (2023); Roy & Etemad (2023) to automate prompt optimization in a few-shot setting. For instance, AnomalyCLIP Zhou et al. (2024) introduced object-agnostic prompts, simplifying prompt crafting while utilizing general anomalous patterns. To address the patch-level misalignment, some methods fine-tune the vision encoder Qu et al. (2024); Chen et al. (2024b); Li et al. (2025); Zhu & Pang (2024); Chen et al. (2023), keep the vision encoder frozen yet further refine its attention modules Li et al. (2024); Zhou et al. (2024), or use deep token tuning in both text and vision encoder Cao et al. (2024). AnomalyCLIP Zhou et al. (2024) follows the second approach, using extra "VV" attention introduced in CLIPSurgery Li et al. (2023) to leverage patch embedding correlations, enhancing CLIP vision encoder segmentation ability. AACLIP Cao et al. (2024) and AdaCLIP Cao et al. (2024) follow the latter approach by jointly tuning the vision and text encoders. AdaCLIP enhances feature alignment by applying k-means clustering to dense visual features and adding learnable linear projection heads to the vision encoder, while AACLIP performs step-by-step fine-tuning of the text and vision encoders. Although effective, our experiments reveal that AnomalyCLIP struggles with image-level generalization, whereas AdaCLIP and AACLIP underperform at pixel-level detection. To address these shortcomings, we propose a context-guided prompt learning strategy to enhance alignment between textual and visual features and extend the technique which proposed by CLIPSurgery, We further improve spatial alignment using models such as DINOv2 motivated by Shi et al. (2024); Lan et al. (2024b). Unlike AdaCLIP, we do not fine-tune the vision encoder, as this can degrade its performance Zhai et al. (2022), and we avoid clustering techniques such as K-Means, which require additional hyperparameter tuning.

## Problem Statement

Let $M_\theta$ denote a pretrained vision-language model (e.g., CLIP) with fixed parameters $\theta$. We consider source anomaly detection datasets $D_{\text{train}}$ from selected domains, where each image $x \in D_{\text{train}} \subset \mathbb{R}^{C \times H \times W}$ is paired with an image-level and pixel-level labels

$$y \in \{0,1\}, \quad S \in \{0,1\}^{H \times W},$$

where $y = 1$ indicating an anomaly and $y = 0$ a normal sample and pixels labeled as 1 marking anomalous regions.

In zero-shot anomaly detection framework, given a prompt $P$ the model produces two continuous anomaly scores for each image $x$:

$$\hat{y}, \hat{S} = M_\theta(x, P),$$

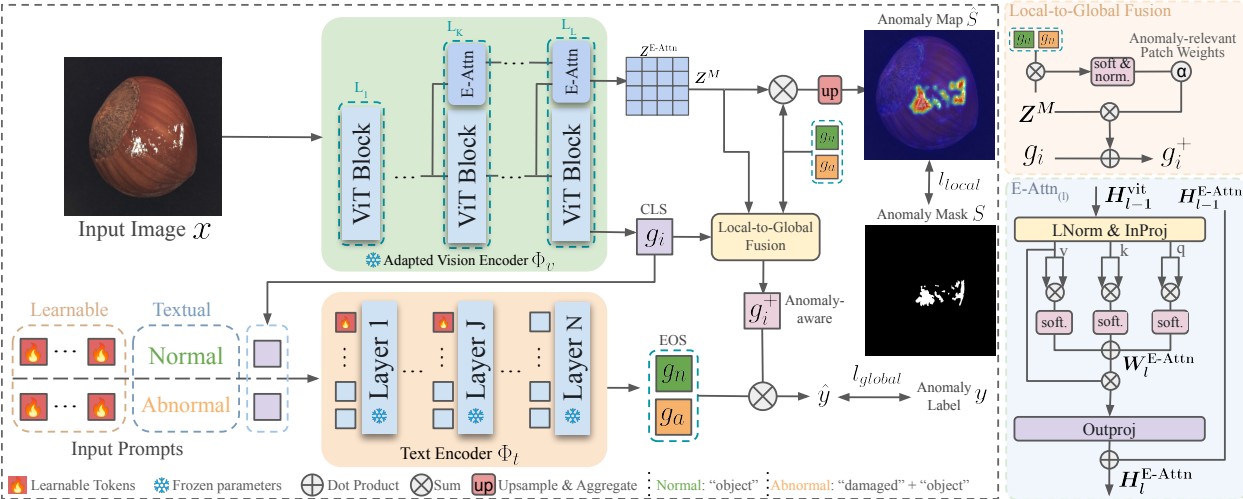

Figure 2: **Crane framework for Zero-shot Anomaly Detection.** First, we extract global embedding $g_i$ and spatially aligned local embeddings $Z^M$ by passing the image through the CLIP's adapted vision encoder $\Phi_v$. Next, we guide learnable prompts with global image context to enhance the capture of fine-grained anomalous patterns. We then compute anomaly map $\hat{S}$ by measuring the similarity between visual embeddings and textual normal/anomalous embeddings. To boost image-level sensitivity to anomalies, we refine the global embedding by incorporating local embeddings weighted by their scores ($g_i^+$) and finally obtain anomaly score $\hat{y}$.

where $\hat{y} \in [0, 1]$ is the image-level anomaly score and $\hat{S} \in [0, 1]^{H \times W}$ is the pixel-level anomaly map. Here, $P$ comprises textual templates (e.g., "a photo of normal CLS") optionally augmented with learnable parameters, which are either prepended to or integrated within the textual input. The final image-level decision—classifying an image as normal or abnormal—is then obtained by thresholding $\hat{y}$ as follows:

$$y' = \begin{cases} 1, & \text{if } \hat{y} > \tau, \\ 0, & \text{otherwise,} \end{cases}$$

where $y' = 1$ denotes an anomalous sample.

A common trend in zero-shot anomaly detection is to optimize the prompt $P$ on $D_{\text{train}}$ while keeping $\theta$ fixed, so that $P$ captures generalizable anomalous features. The optimized prompt $P^*$ is then applied to new domains—where labeled anomaly data is unavailable—for both image- and pixel-level detection.

## 3 Method

We propose a unified framework that leverages CLIP as a zero-shot backbone ($M_\theta$) for classification and segmentation, while adapting it for anomaly detection to bridge the domain gap between CLIP's pretraining and specialized detection tasks. As shown in Figure 2, we enhance the vision encoder with an E-Attn module that refines attention weights to better capture localized anomalous cues. We also learn class-agnostic input prompts ($P$) and trainable tokens in the text encoder ($\Phi_t$), guided by visual feedback from the vision encoder ($\Phi_v$). Additionally, an anomaly-aware local-to-global fusion mechanism integrates dense anomalous features into the global embedding, yielding more anomaly-sensitive representations for robust zero-shot generalization at both image and pixel levels across unseen domains.

### 3.1 Local and Global Visual Feature Extraction

To craft the image anomaly score, $\hat{y}$, and the corresponding anomaly localization map, $\hat{S}$, in zero-shot, each input image needs to be modeled at both local (dense) and global (image-level) through our vision encoder

$\Phi_v$ and then compared with the normal/abnormal text features extracted by the text encoder $\Phi_t$. Each component is explained in detail below.

**Adapted CLIP Vision Encoder ($\Phi_v$).** Given an input image $x$ the vision encoder produces two outputs: a global embeddings $g_i \in \mathbb{R}^D$, used for classification, and local embeddings $Z^{\mathrm{M}} \in \mathbb{R}^{N \times D}$ used for segmentation, where $D$ denotes the dimension of embeddings and $N$ Number of patch embeddings which are obtained from the *E-Attn* branch. The global representation $g_i$ is the [CLS] token from the forward pass of the CLIP vision transformer, which extracts textually aligned image-level features. However, for local embeddings, we opt not to use the ViT's original embeddings, as the emphasis on global image-text alignment during CLIP's pretraining has led to degraded similarity between corresponding patch embeddings across layers, resulting in inaccurate segmentation Li et al. (2023); Lan et al. (2024a).

To address this, we adapt CLIP by replacing the Query-Key-based ($QK^T$) attention weighting with a self-correlation weighting scheme to reinforce semantic correlation across layers. Given $E \in \mathbb{R}^{N \times D}$ as a set of $N$ embeddings, a self-correlation attention weighting can be defined as:

$$A(E) = \mathrm{softmax}\Big(\frac{EE^\top}{\sqrt{D}}\Big), \quad A(E) \in \mathbb{R}^{N \times N},$$

where $A(E)$ is a weight matrix that captures pairwise similarities between embeddings. We extend self-correlation weighting by applying the *Extended Self-Correlation Attention (E-Attn)* branch at layer $l$ of CLIP ViT. Given $K_l, Q_l, V_l \in \mathbb{R}^{N \times D}$ as the *key*, *query*, and *value*, we compute attention weights as follows:

$$W_l^{\mathrm{E\text{-}Attn}} = A\big(K_l\big) + A\big(Q_l\big) + A\big(V_l\big),$$

Then, the same as the standard attention Dosovitskiy et al. (2020), $W_l^{\mathrm{E\text{-}Attn}}$ is used to aggregate $V_l$ tokens, producing $H_l^{\mathrm{E\text{-}Attn}}$ as the attention output. Since early layers have less expressive representations, this branch operates on the last $K$ layers of the ViT. Given $L$ as the total number of layers, for input image $x$ the final output $Z^{\mathrm{M}}$ is obtained by aggregating intermediate outputs:

$$Z^{\mathrm{M}} = Z^{\mathrm{E\text{-}Attn}} = \sum_{l=L-K+1}^{L} H_l^{\mathrm{E\text{-}Attn}}, \quad Z^{\mathrm{E\text{-}Attn}} \in \mathbb{R}^{N \times D}.$$

**Text Encoder ($\Phi_t$).** To obtain the optimum textual alignment with anomalous visual embeddings, we employ prompt learning and deep token tuning within the text encoder. The standard transformer block $t_l(.)$ at layer $l$ is defined as:

$$[\mathrm{SOS}_l, H_l, \mathrm{EOS}_l] = t_l\big([\mathrm{SOS}_{(l-1)}, H_{(l-1)}, \mathrm{EOS}_{(l-1)}]\big)$$

where $[\mathrm{SOS}_l]$ and $[\mathrm{EOS}_l]$ are special tokens marking the sequence's start and end, and $H_l$ represents intermediate token representations. For each layer $l = 2..J+1$, we replace the first $M$ tokens of $H_{(l-1)}$ with learnable tokens $\tau_l \in \mathbb{R}^{M \times D}$ to capture anomaly-specific knowledge. At the final layer, the [EOS] token, which aggregates the semantic representation of the input prompt, is used as the textual feature.

To design the input prompt, we abandon class-based, manually crafted templates in favor of object-agnostic learnable prompts Zhou et al. (2024). As a result, we learn only two prompts (normal and anomalous), instead of two per dataset category. This approach leverages the shared structural patterns of anomalies across domains and reduces the need for domain-specific prompt engineering. To achieve this, we use a set of $E$ learnable tokens for each of the normal and anomalous prompts, denoted by $\tau^n, \tau^a \in \mathbb{R}^{E \times D}$, and concatenate them with general product and state textual descriptions: "object" for the normal case and "damaged object" for the anomalous case, appended after the learnable sets.

Additionally, we introduce *context-guided prompt learning*, which integrates the image-level representation $g_i$ into textual prompts during training, enabling the model to better capture fine-grained distributions. In summary, the normal $g_n$ and abnormal textual embedding $g_a$ are constructed as:

$$g_n = T\Big([\tau_1^n, \tau_2^n, \ldots, \tau_E^n, \text{``object''}, g_i]\Big),$$

$$g_a = T\Big([\tau_1^a, \tau_2^a, \ldots, \tau_E^a, \text{``damaged''}, \text{``object''}, g_i]\Big).$$

**Calculating anomaly likelihood.** Having obtained the textual $G = \{g_n, g_a\}$ and visual embeddings $e \in \{g_i, Z_{j,k}\}$, where $Z_{j,k}$ represents patch embedding at the position $(j, k)$ of unflattened local feature branch $Z \in Z^M = \{Z^{\text{D-Attn}}, Z^{\text{E-Attn}}\}$ for input image $x$, we can now compute the likelihood of each visual embedding belonging to the anomalous class $(p_a)$ by applying the Softmax function to similarity scores:

$$p_a(e, G) = \frac{\exp\left(\langle e, g_a \rangle / \tau\right)}{\exp\left(\langle e, g_a \rangle / \tau\right) + \exp\left(\langle e, g_n \rangle / \tau\right)} \tag{1}$$

where the temperature $\tau$ is set to 100 according to CLIP's hyperparameter details Radford et al. (2021). We denote the probability of a visual embedding $e$ being abnormal, $p_a(e, G)$, as its anomaly score.

**Anomaly-aware Local-to-Global Fusion** The image-level representation $g_i$, trained to capture a global representation of an image, may fail to encode the discriminative fine-grained anomalous cues due to its global focus. To address this issue, we propose a *score-based spatial pooling* mechanism that fuses patch-level features into $g_i$ based on their anomaly scores, ensuring the final representation preserves global semantics while capturing anomaly cues. For each local feature map $Z \in \mathbb{R}^{H \times W \times D}$ from $Z^M$, The anomaly-aware representation is constructed as:

$$g_a^z = \frac{\sum_{j,k} p_a(Z_{j,k}, G) Z_{j,k}}{\sum_{j,k} p_a(Z_{j,k}, G)}, \quad Z \in Z^M.$$

After obtaining $g_a^{\text{D-Attn}}$ and $g_a^{\text{E-Attn}}$ for each branch, we fuse these anomaly-aware embeddings into $g_i$ via averaging:

$$\bar{g}_a = \frac{g_a^{\text{E-Attn}} + g_a^{\text{D-Attn}}}{2}, \quad g_i^+ = \frac{g_i + \bar{g}_a}{2},$$

where $g_i^+$ is the anomaly-aware global representation used for anomaly classification.

`Crane`$^+$ **for enhanced localization.** Domains such as medical imaging present unique challenges for anomaly detection, where anomalies often differ from normal tissue in texture or intensity by only a few pixels—requiring highly detailed feature discrimination. To address this, we propose a module that adapts DINOv2 to our zero-shot setting for more robust localization . While DINOv2 encodes spatially sensitive patch-level features, it lacks the semantic required for zero-shot capabilities. We integrate it through a module called *DINO-guided spatial attention (D-Attn)*, which—like E-Attn—replaces the standard attention weights with spatially aware ones, as described:

$$Z^{\text{D}} = f_{\text{D}}(x), \quad S = \langle Z^{\text{D}}, Z^{\text{D}} \rangle,$$

$$W^{\text{D-Attn}} = \text{softmax}(S + M), \quad M_{ij} = \begin{cases} 0, & S_{ij} \geq \epsilon, \\ -\infty, & S_{ij} < \epsilon, \end{cases}$$

where $Z^{\text{D}} \in \mathbb{R}^{N \times D}$ is DINOv2's output patch embeddings, the operator $\langle ., . \rangle$ is cosine similarity and $S \in \mathbb{R}^{N \times N}$ is the computed similarity matrix. To refine $S$, we apply a masking mechanism to discard low similarity scores, followed by softmax normalization over the last dimension. The refined similarity matrix $W^{\text{D-Attn}}$ is then used at the final layer $L$ as attention weights, producing attention output $H_L^{\text{D-Attn}} \in \mathbb{R}^{N \times D}$. In `Crane`$^+$ this output is then concatenated to $Z^{\text{E-Attn}}$ yielding $Z^{\text{M}}$. Notably, this module can be implemented efficiently, as DINOv2 features are extracted in parallel with CLIP. Detailed evaluation available in computational analysis.

## 3.2 Training

To train the text encoder $\Phi_t$, we employ global and local loss functions. For input image $x$ the global loss ensures alignment between the global embedding $g_i^+$ and its corresponding textual embedding ($g_a$ and $g_n$) being learned using Binary Cross Entropy and image-level label $y$:

$$L_{\text{global}} = \text{BCE}\left(y, p_a(g_i, G)\right).$$

For local loss, we use Focal Lin et al. (2017) and Dice Sudre et al. (2017) loss at pixel-level. For output feature map $Z \in Z^M$, we compute normal and anomaly maps $S_n^Z$ and $S_a^Z$ based on the equation 1, then apply bilinear upsampling $\text{Up}(\cdot)$ to match the anomaly mask $S \in \mathbb{R}^{H \times W}$:

$$L_{\text{local}}^Z = \text{Focal}\left(\text{Up}([S_n^Z, S_a^Z]), S\right) + \text{Dice}\left(\text{Up}([S_n^Z, S_a^Z]), S\right).$$

Finally, we combine both terms using $\lambda$ as a weighting factor controlling the contribution of the local loss.

$$L_{\text{total}} = L_{\text{global}} + \lambda \sum_{Z \in Z^M} L_{\text{local}}^Z.$$

## 4 Inference

For each input image $x$, after computing visual outputs including local features $Z \in Z^M$ and anomaly-aware global embedding $g_i^+$ alongside textual embeddings $G = \{g_a, g_n\}$ in forward pass, anomaly score $\hat{y}$ and low-resolution anomaly maps $\hat{S}_a^Z$ is calculated as follows:

$$\hat{y} = p_a(g_i^+, G), \quad \hat{S}_{a,(j,k)}^Z = p_a(Z_{j,k}, G),$$

where $\hat{S}_{a,(j,k)}^Z$ is anomaly score for each patch at location $(j,k)$. We then perform averaging on $\hat{S}_a^{\text{D-Attn}}$ and $\hat{S}_a^{\text{E-Attn}}$, then apply bilinear upsampling and Gaussian filter smoothing to obtain final anomaly maps $\hat{S}$.

## 5 Experiments

### 5.1 Experiments Setting

**Datasets.** To ensure a comprehensive evaluation, we conduct experiments on 14 real-world anomaly detection datasets spanning industrial defect detection and medical abnormality analysis across diverse domains and anomaly types. For industrial anomaly detection, we utilize MVTec AD Bergmann et al. (2019), VisA Zou et al. (2022), DTD-Synthetic Aota et al. (2023), SDD Tabernik et al. (2020), BTAD Mishra et al. (2021), DAGM Wieler & Hahn (2007), and MPDD Jezek et al. (2021), which primarily focus on texture and structural defects in manufactured scenarios. For medical anomaly detection, we evaluate on ISIC Gutman et al. (2016), CVC-ColonDB Tajbakhsh et al. (2015), CVC-ClinicDB Bernal et al. (2015), TN3K Gong et al. (2021), BrainMRI Kanade & Gumaste (2015), HeadCT Kitamura (2018), and BR35H Hamada (2020), covering pathological abnormalities across dermoscopic, colonoscopic, retinal, and brain imaging domains. Following prior works Zhou et al. (2024); Cao et al. (2024); Chen et al. (2023), for all experiments, we train our model on test set of MVTec-AD and evaluate its generalization to other datasets. To assess performance on MVTec-AD itself, we train the model on test set of VisA, which contains non-overlapping sample categories. Further details on preprocessing and hyperparameters are provided in Appendix A.

**Evaluation Metrics.** Following previous studies Jeong et al. (2023); Deng et al. (2023), we employ AUROC, AP, and F1-max to evaluate the model's ability to distinguish between normal and anomalous images. For pixel-level anomaly localization, we use AUROC, AUPRO, and F1-max to assess effectiveness in identifying anomalous regions. For each dataset, we report the average performance across object categories as the dataset-level result.

### 5.2 Main Results

**Baselines.** We compare our method against two categories of CLIP-based zero-shot anomaly detection (ZSAD) approaches:training-free methods including AnoVL Deng et al. (2023) and training-required methods which involve training on an auxiliary dataset before inference, such as VAND Chen et al. (2023), AnomalyCLIP Zhou et al. (2024), AdaCLIP Cao et al. (2024), and AA-CLIP Ma et al. (2025). Details on the performance values reported for each model are provided in Appendix C.

Table 1: **Comparison of ZSAD methods across industrial and medical domains.** We compare our method against the state-of-the-art across 14 diverse industrial and medical datasets. The best performance is highlighted in **bold**, while the second-best is underlined. A † symbol next to a method name indicates training-free models. On industrial benchmarks, Unlike AnomalyCLIP and AdaCLIP, which fail to achieve consistent improvements, both versions of our model enhance the state-of-the-art in image-level and pixel-level metrics. On medical benchmarks, Our method achieves competitive performance or significant improvement compared to the SOTA as shown by image and pixel metrics.

| Metric | Dataset | AnoVL† | VAND | AnomalyCLIP | AdaCLIP | AA-CLIP | Ours (Crane) | Ours (Crane⁺) |
|---|---|---|---|---|---|---|---|---|
| Image-level ↑ (AUROC, AP, F1-max) | MVTec | (92.5, 95.1, 93.2) | (86.1, 93.5, 88.9) | (91.5, 96.2, 92.7) | (89.2, 95.7, 90.6) | (90.5, 95.4, 90.7) | (93.8, 97.5, 93.8) | (93.9, 97.6, 93.6) |
| | VisA | (79.2, 80.2, 79.7) | (78.0, 81.4, 80.7) | (82.1, 85.4, 80.4) | (85.8, 79.0, 83.1) | (84.6, 82.9, 78.8) | (85.3, 87.9, 82.6) | (83.6, 86.7, 81.2) |
| | MPDD | (72.7, 83.6, 88.3) | (73.0, 80.2, 76.0) | (77.0, 82.0, 80.4) | (76.0, 80.2, 82.5) | (74.7, 76.8, 79.7) | (81.4, 84.8, 81.6) | (81.0, 84.1, 83.0) |
| | BTAD | (80.3, 72.8, 73.0) | (73.6, 68.6, 82.0) | (88.3, 87.3, 83.8) | (88.6, 93.8, 88.2) | (94.8, 97.3, 93.7) | (94.4, 95.9, 91.7) | (96.3, 97.0, 93.7) |
| | KSDD | (94.4, 90.8, 88.0) | (79.8, 71.4, 85.2) | (84.7, 80.0, 82.7) | (97.1, 89.6, 90.7) | (86.8, 59.5, 55.9) | (97.8, 94.3, 91.6) | (97.8, 94.5, 89.7) |
| | DAGM | (89.7, 76.1, 74.7) | (94.4, 83.8, 91.8) | (97.5, 92.3, 90.1) | (99.1, 88.5, 97.5) | (95.0, 85.3, 82.6) | (99.2, 97.4, 95.8) | (98.9, 96.1, 94.7) |
| | DTD | (94.9, 93.3, 97.3) | (94.6, 95.0, 96.8) | (93.5, 97.0, 93.6) | (95.5, 97.3, 94.7) | (91.3, 97.0, 92.8) | (96.3, 98.5, 95.3) | (95.8, 98.2, 94.6) |
| | Average | (86.2, 84.6, 84.9) | (81.6, 82.0, 85.9) | (87.8, 88.6, 87.2) | (90.2, 89.2, 89.6) | (88.3, 85.2, 81.9) | (92.6, 93.7, 90.3) | (92.5, 93.5, 90.1) |
| | HeadCT | (82.3, 81.2, 79.1) | (89.2, 89.5, 82.1) | (93.4, 91.6, 90.8) | (91.8, 90.6, 84.1) | (95.2, 92.0, 90.1) | (95.3, 95.7, 91.1) | (94.6, 95.4, 89.7) |
| | BrainMRI | (84.3, 89.2, 84.8) | (89.6, 91.0, 88.5) | (90.3, 92.2, 90.2) | (93.5, 95.6, 89.7) | (93.0, 92.1, 91.2) | (95.4, 96.1, 93.9) | (96.3, 97.4, 93.5) |
| | Br35H | (80.0, 80.7, 75.2) | (91.4, 91.9, 84.2) | (94.6, 94.7, 89.1) | (92.3, 93.2, 85.3) | (96.0, 94.0, 90.9) | (96.3, 96.8, 91.7) | (96.4, 97.2, 91.0) |
| | Average | (82.2, 83.7, 79.7) | (90.1, 90.8, 84.9) | (92.8, 92.8, 90.0) | (92.5, 93.1, 86.4) | (94.7, 92.7, 90.7) | (95.7, 96.2, 92.2) | (95.7, 96.7, 91.4) |
| Pixel-level ↑ (AUROC, AUPRO, F1-max) | MVTec | (90.6, 77.8, 36.5) | (87.6, 44.0, 39.8) | (91.1, 81.4, 39.1) | (88.7, 37.8, 43.4) | (91.9, 45.9, 47.0) | (91.3, 84.6, 41.3) | (91.2, 88.1, 43.8) |
| | VisA | (85.2, 60.5, 14.6) | (94.2, 86.8, 32.3) | (95.5, 87.0, 28.3) | (95.5, 72.9, 37.7) | (95.5, 25.0, 30.5) | (95.1, 87.5, 30.9) | (95.3, 90.6, 30.2) |
| | MPDD | (62.3, 38.3, 15.6) | (94.1, 83.2, 30.6) | (96.5, 88.7, 34.2) | (96.1, 62.8, 34.9) | (96.7, 26.1, 28.9) | (97.0, 89.3, 38.2) | (97.6, 93.2, 42.0) |
| | BTAD | (75.2, 40.9, 23.4) | (60.8, 25.0, 38.4) | (94.2, 74.8, 49.7) | (92.1, 20.3, 51.7) | (97.0, 54.3, 55.6) | (96.6, 81.3, 56.9) | (96.7, 86.8, 61.1) |
| | KSDD | (97.1, 82.6, 23.1) | (79.8, 65.1, 56.2) | (90.6, 67.8, 51.3) | (97.7, 33.8, 54.5) | (98.1, 29.6, 37.4) | (97.9, 95.9, 60.2) | (99.2, 97.4, 62.4) |
| | DAGM | (79.7, 56.0, 12.8) | (82.4, 66.2, 57.9) | (95.6, 91.0, 58.9) | (91.5, 50.6, 57.5) | (95.8, 51.5, 53.7) | (96.3, 91.2, 67.2) | (96.2, 93.8, 66.8) |
| | DTD | (97.7, 90.5, 46.8) | (95.3, 86.9, 72.7) | (97.9, 92.3, 62.2) | (97.9, 72.9, 71.6) | (96.5, 61.4, 60.5) | (98.3, 93.3, 68.8) | (98.8, 96.0, 71.8) |
| | Average | (84.0, 63.8, 24.7) | (84.9, 65.3, 46.8) | (94.5, 83.3, 46.2) | (94.2, 50.1, 50.2) | (95.9, 42.0, 44.8) | (96.1, 89.0, 51.9) | (96.4, 92.3, 54.0) |
| | ISIC | (92.6, 82.2, 76.6) | (89.5, 77.8, 71.5) | (89.7, 78.4, 70.6) | (90.3, 54.7, 72.6) | (93.8, 85.3, 78.4) | (88.1, 75.3, 69.8) | (90.6, 83.2, 73.4) |
| | ColonDB | (76.2, 44.1, 26.8) | (78.4, 64.6, 29.7) | (81.9, 71.3, 37.3) | (82.6, 66.0, 36.1) | (84.0, 33.4, 39.0) | (82.5, 73.0, 36.0) | (86.0, 78.6, 40.2) |
| | ClinicDB | (79.7, 51.4, 36.3) | (80.5, 60.7, 38.7) | (82.9, 67.8, 42.1) | (82.8, 66.4, 40.9) | (89.9, 54.0, 53.8) | (84.0, 69.3, 42.5) | (88.3, 74.5, 47.9) |
| | TN3K | (70.2, 34.4, 32.3) | (73.6, 37.8, 35.6) | (81.5, 50.4, 47.9) | (76.8, 34.0, 40.7) | (80.4, 39.4, 42.5) | (79.4, 48.8, 44.7) | (80.4, 51.7, 45.5) |
| | Average | (79.7, 53.0, 43.0) | (80.5, 60.2, 43.9) | (84.0, 67.0, 49.5) | (83.1, 55.3, 47.6) | (87.0, 53.0, 53.4) | (83.5, 66.6, 48.2) | (86.4, 72.0, 51.8) |

**Zero-shot Performance over Industrial Datasets.** We evaluate the generalization of `Crane` on seven industrial datasets, as shown in Table 1. We report results for both the base model `Crane` and enhanced model `Crane⁺`. The industrial image-level results show that both versions outperform prior works by a similar margin. Notably, `Crane` improves the state-of-the-art by 2.4% in AUROC, 4.5% in AP, and 0.7% in F1-max on average. These gains highlight the effectiveness of our contributions designed to address the limited sensitivity of global features to local anomalous cues—namely, context-guided prompt learning and anomaly-aware local-to-global fusion.

In the industrial pixel-level results, we observe that `Crane` demonstrates consistent and remarkable improvements, achieving 0.2% higher AUROC, 5.7% higher AUPRO, and a 1.7% increase in F1-max on average. `Crane⁺` pushes the margin even further, improving AUROC by 0.5%, AUPRO by 9.0%, and F1-max by 3.8%, demonstrating that DINOv2 injects complementary knowledge beyond CLIP vision encoder via D-Attn modules. These improvements are attributed to the architectural modifications designed to preserve spatial alignment—namely, the E-Attn branch in both `Crane` and `Crane⁺`, and the complementary D-Attn in `Crane⁺`.

**Zero-shot Generalization to Medical Domain.** Here, we assess how well the features learned on industrial datasets generalize to domains far from the industrial domain. To this end, we evaluate `Crane` on seven medical datasets spanning diverse applications, including skin cancer detection in photography images, colon polyp identification in endoscopy images, thyroid nodule detection in ultrasound images, and brain tumor detection in MRI images. The goal is to determine whether the model has developed a broader understanding of normality and abnormality.

Table 1 presents the results, showing a consistent image-level trend: both versions outperform prior SOTA by a similar margin, improving average performance by 1.0% to 3.6% across reported metrics. At the pixel level on medical datasets, however, the gains are more mixed: `Crane` trails AA-CLIP in AUROC and F1-max by 3.5% and 5%, respectively, while substantially outperforming it by 13.6% in AUPRO, a more precise metric that is also more commonly used in practice Bergmann et al. (2020). With `Crane⁺`, the gaps in AUROC and F1-max shrink to a subtle margin (0.6% and 1.6%), while it outperforms AA-CLIP even further by 19% in

Table 2: **Ablation analysis of key components.** Performance is reported at image-level (I-ROC, I-F1-max) and pixel-level (P-ROC, P-F1-max) on MVTec-AD and VisA. Higher values indicate improved performance. Default configuration is colored.

(a) **Effect of anomaly-aware local-to-global feature fusion.**

| | Image-level | |
| Score | **MVAD** | **VisA** |
| --- | --- | --- |
| ✗ | (90.7, 92.3) | (79.2, 78.9) |
| ✓ | (**94.7**, **94.3**) | (**82.6**, **80.6**) |

(b) **Effect of Context-guided Prompt Learning.**

| | Pixel-level | | Image-level | |
| Train | **MVAD** | **VisA** | **MVAD** | **VisA** |
| --- | --- | --- | --- | --- |
| ✗ | (91.7, 43.5) | (94.8, 28.1) | (94.0, 93.3) | (81.9, 80.2) |
| ✓ | (**92.1**, **44.7**) | (**95.5**, **29.2**) | (**94.7**, **94.3**) | (**82.6**, **80.6**) |

(c) **Effect of different Self-correlations for E-Attn.**

| | Pixel-level | | Image-level | |
| Self-cors. | **MVAD** | **VisA** | **MVAD** | **VisA** |
| --- | --- | --- | --- | --- |
| kk | (91.6, 43.5) | (94.9, 27.9) | (93.6, 93.7) | (82.2, 80.4) |
| vv | (92.1, 43.8) | (95.0, 26.7) | (93.8, 93.3) | (82.4, **80.7**) |
| qq | (91.8, 43.9) | (95.0, **29.4**) | (94.1, **94.0**) | (80.2, 79.4) |
| qq+kk | (91.6, 44.0) | (95.2, 29.3) | (94.0, 93.9) | (82.3, 80.5) |
| qq+kk+vv | (**92.1**, **44.7**) | (**95.5**, 29.2) | (**94.7**, **94.3**) | (**82.6**, 80.6) |

(d) **Effect of D-Attn.** Superscripts 1, 2, and 3 refer to the specific models used: (1) CLIP-L14, (2) DINOv1-B8, (3) DINOv2-B14.

| | Pixel-level | | Image-level | |
| Branch | **MVAD** | **VisA** | **MVAD** | **VisA** |
| --- | --- | --- | --- | --- |
| E-Attn[1] | (91.2, 40.8) | (94.3, **29.5**) | (93.7, 93.9) | (**82.7**, 80.4) |
| D-Attn[2] | (91.5, 43.7) | (94.0, 27.1) | (91.1, 92.9) | (78.4, 78.9) |
| D-Attn[3] | (92.0, 42.7) | (93.7, 26.7) | (91.6, 92.6) | (78.3, 78.3) |
| Both[1+3] | (**92.1**, **44.7**) | (**95.5**, 29.2) | (**94.7**, **94.3**) | (82.6, **80.6**) |

AUPRO, demonstrating strong zero-shot cross-domain generalization and highlighting the practical benefits of the proposed methods over previous methods in real-world applications.

## 5.3 Ablation Study

To assess the effectiveness of each contribution, we conduct a series of controlled ablation experiments. All models are trained for five epochs, with only a single component modified at each stage while keeping all other factors unchanged. We report F1-max and AUROC scores on VisA and MVTec-AD for both image- and pixel-level performance.

**Anomaly-aware Local-to-Global Fusion.** A purely global image representation from the CLIP image encoder can be too coarse, as CLIP is trained to align images with relatively coarse text descriptions Salehi et al. (2023); Oquab et al. (2023). In our anomaly detection setting, the distributions of normal and abnormal inputs can be very close, as illustrated in Figures 4, 5, and 8 of the appendix, which leads to highly similar global representations. To address this limitation, we fuse local dense features containing abnormal regions with the global representation, enabling the model to learn more discriminative prompts. Moreover, since the informative cues may be confined to only a few dense patches while large parts of the image are irrelevant or distracting, we first score the patches and then fuse only the most relevant ones. This motivates our combination of global and local information: the global representation preserves the overall semantic context of the image, while the local features allow the model to focus on fine-grained regions where the critical evidence appears. In this way, the two representations play complementary roles. As shown in Table 3.a, aggregating anomalous local embeddings into the global `[CLS]` token improves F1-max by 1.9% on average across VisA and MVTec-AD datasets.

**Context-guided Prompt Learning.** Rather than learning a single shared set of normal and abnormal prompts for all inputs, the model conditions the prompts on the visual context of each image. This produces text representations that are better aligned with the corresponding input features, enabling the model to learn normal and abnormal prompts in an instance-aware manner. By making the prompts input-dependent, the method reduces reliance on dataset-specific prompt biases and improves generalization across visually diverse samples. To assess the impact of context-guided prompt learning, we compare model performance with and without concatenating $g_i$ (the visual [CLS] token) to the input prompts. As shown in Table 3.b, incorporating train-domain representations during training consistently enhances both pixel-level and image-level performance. On average, F1-max increases by 0.9% across both performance levels. This

improvement can be attributed to the ability to better model fine-grained anomalous patterns given the global context of train images.

**Extended Self-Correlation Attention.** In Table 3.c the effect of using separate attention tokens ($qq$, $kk$, or $vv$) and their combinations ($qq + kk$ or $qq + kk + vv$) for self-correlation attention is examined. Extending self-correlation to include all three tokens ($qq + kk + vv$) improves pixel-level F1-max by 1.7% and image-level F1-max by 0.5% over the $vv$-only baseline. These results suggest that $qq$, $kk$, and $vv$ capture certain spatial information exclusively, therefore combining them improves alignment.

**DINO-guided Spatial Attention.** Table 3.d investigates the effectiveness of utilizing D-attn by measuring the performance without (1st row) and with the module (4th row). We also compare different variants of DINO. Our adapted CLIP model (1st row) outperforms standalone D-Attn (2nd and 3rd rows) in image classification by a significant margin. Utilizing D-attn along with E-attn (4th row) improves pixel-level F1-max by 1.8% and image-level F1-max by 0.3% on average compared to the first row. This improvement highlights the complementary strengths of the two branches: D-Attn excels at capturing visual similarity across patches but lacks semantic information, whereas E-Attn better preserves semantic information but struggles with aligning more fine-grained details. As also supported by the qualitative comparisons in Figure 3, this attention design helps suppress distracting regions and produces features that better highlight abnormal areas. For further evaluation on using SAM Kirillov et al. (2023) in D-attn, refer to Appendix F.

**Computational Analysis**

We evaluate the computational efficiency of `Crane` against competing methods in terms of training time, inference throughput, and memory usage (Table 3) using a single A6000 GPU. AA-CLIP achieves the highest inference FPS; however, it lags substantially behind other models in mean industrial performance. Among the rest, both variants of our method operate with comparable or lower computational cost than others, while achieving large performance gains, indicating a better performance–efficiency trade-off. The modest impact on inference speed in (`Crane`$^+$) stems from a parallel design in which DINOv2 features are extracted concurrently with CLIP, avoiding additional sequential overhead.

Table 3: **Computational efficiency comparison.** For each model, we report training time per epoch, inference throughput (FPS), and inference memory on MVTec-AD, along with mean pixel-level AUPRO and image-level AP over industrial datasets.

| Model | Train (min/epc) | Infer. (FPS) | Mem. (GB) | Mean Perf. (P-AUPRO, I-AP) |
|---|---|---|---|---|
| AdaCLIP | 9.36 | 3.43 | 4.21 | 50.1, 89.2 |
| AA-CLIP | 7.55 | **5.12** | 5.19 | 42.0, 85.2 |
| AnomalyCLIP | **7.21** | 3.99 | **3.76** | 83.3, 88.6 |
| Crane | 7.38 | 3.67 | 3.81 | 89.0, **93.7** |
| (Crane$^+$) | 8.15 | 3.38 | 4.20 | **92.3**, 93.5 |

## 6 Visual Analysis

To provide an intuitive comparison, we present anomaly maps generated by the top competing models: VAND, AdaCLIP, AA-CLIP, AnomalyCLIP, and both versions of our model, `Crane` and `Crane`$^+$, across industrial images from MVTec-AD, VisA, MPDD, BTAD, DAGM, and DTD-Synthetic, as well as medical images from ISIC, CVC-ClinicDB, and BrainMRI. Note that BrainMRI only has image-level labels, and the provided sample is annotated by a medical professional. As shown in the Figure 3, VAND suffers from high false positive rates (FPR), while AdaCLIP exhibits a high false negative rate (FNR). AA-CLIP and AnomalyCLIP improve sensitivity by reducing FNR but still struggle with high FPR. By leveraging stronger semantic correlations among patches, `Crane` reduces both FPR and FNR over previous methods, yielding

tidier and more precise results. This effect is further enhanced in `Crane`$^+$, demonstrating superior localization performance.

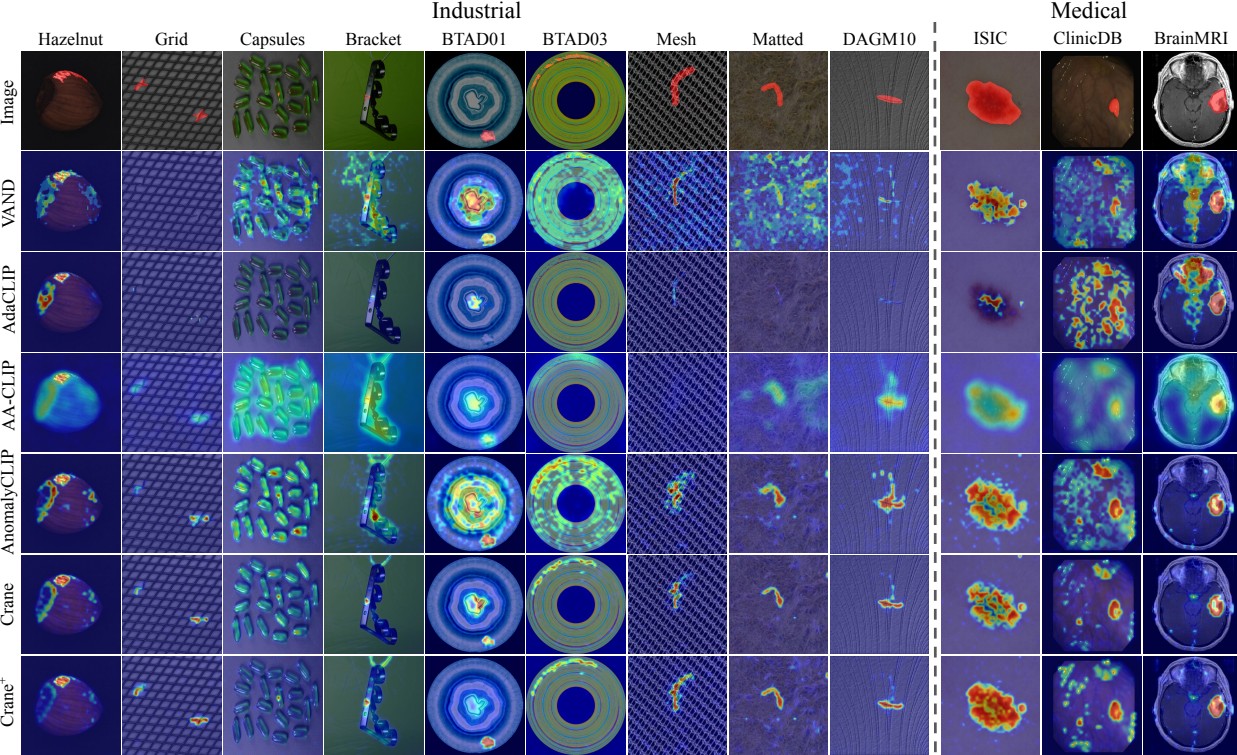

Figure 3: **Qualitative localization comparison**. Models are trained on industrial datasets as described in section 5.1. By leveraging stronger semantic correlations among patches, `Crane` achieves remarkable reductions in both false positive and false negative rates compared to prior methods, resulting in cleaner and more precise anomaly localization. This advantage is further amplified in `Crane`$^+$, demonstrating state-of-the-art localization performance.

# 7    Conclusion

In this work, we introduced `Crane` for zero-shot anomaly detection. Our approach addresses major challenges of CLIP in this task, namely the global features' limited sensitivity to local anomalous patterns and spatial misalignment. For the first one, we guide prompts using the global visual token and fuse anomaly-relevant local patches into the global token. For the latter, we propose an extended self-correlation attention mechanism, and for more complex domains, DINO-guided spatial attention with relatively low computational overhead. The extensive experiments across 14 medical and industrial datasets demonstrate the effectiveness of the four components in tackling the two limitations, achieving state-of-the-art performance in zero-shot anomaly detection and localization. More discussion on limitations in Appendix G.2.

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

# Appendix

## A    Implementation Details

In this study, we use the publicly available pre-trained CLIP (ViT-L/14@336px)[1] as the default backbone and pre-trained DINOv2-B14[2] for further spatial alignment. Each input prompt is assigned 12 learnable token embeddings, while 4 learnable deep token embeddings per layer are inserted into the first 9 layers of the text encoder. We use the Adam optimizer with a learning rate of 0.001 and `betas=(0.6, 0.999)`, and train the model once using a batch size of 8. The random seed for all algorithms is set to 111 to ensure reproducibility. Input images to the vision tower are resized to 518×518 and undergo the same normalization during both training and inference. All experiments are conducted using PyTorch 1.13.1 on a single NVIDIA A6000 GPU.

To evaluate zero-shot anomaly detection performance, we train `Crane` on the test set of MVTec-AD and assess its generalization across other datasets. For evaluation on MVTec-AD, the model is trained on the test data of VisA. MVTec-AD and VisA contain distinct object categories with no overlap with the samples in other datasets, and the benchmark datasets used in this study span diverse domains, ensuring minimal category overlap across datasets.

## B    Additional Ablations

### B.1    Incremental Ablations

Here we show the ablation table in an incremental way, starting from the AnomalyCLIP baseline and sequentially adding each proposed component. As shown, context-guided prompt learning improves both image- and pixel-level performance through better input-dependent alignment; extended self-correlation attention mainly improves pixel-level performance by refining spatial features; score-based aggregation primarily boosts image-level performance by enhancing the global representation while leaving dense predictions unchanged; and D-Attn provides the final refinement, especially for localization.

| Module | MVTec AD | | VisA | |
|---|---|---|---|---|
| | Pixel-level | Image-level | Pixel-level | Image-level |
| Baseline (AnomalyCLIP) | (89.8, 36.9) | (91.0, 91.2) | (94.0, 25.2) | (80.1, 78.3) |
| + Context-guided Prompt Learning | (90.6, 38.7) | (92.2, 92.3) | (94.5, 26.7) | (81.0, 79.2) |
| + Extended Self-Correlation Attention | (91.2, 40.8) | (92.2, 92.3) | (94.3, 29.5) | (81.0, 79.2) |
| + Score-based Aggregation | (91.2, 40.8) | (93.7, 93.9) | (94.3, 29.5) | (82.7, 80.4) |
| + D-Attn | (92.1, 44.7) | (94.7, 94.3) | (95.5, 29.2) | (82.6, 80.6) |

Table 4: Illustrative incremental module ablation with AnomalyCLIP as the starting point. Performance is reported at pixel-level (P-ROC, P-F1-max) and image-level (I-ROC, I-F1-max).

### B.2    Simple Anomaly Score Aggregation vs. Score-based Local-to-global Feature Fusion

We further compare the introduced score-based feature aggregation with simple score aggregation. To do so, in `Crane` , we replace the score-based local-to-global feature fusion with score aggregation, in which instead of mean anomaly-aware feature, mean anomaly score of patches is calculated and averaged with image-level anomaly score. As shown in the table 5, aggregating features yields higher performance than simple score aggregation, suggesting that aggregated anomaly score is less effective in capturing anomalous regions that might exist in a picture, as it mainly represents the most frequent regions, especially since the most common patches in a picture are often normal, it will be biased towards normality, reducing precision respectively. Whereas, in feature aggregation the anomalous directions are retained and reflected in the final cosine similarity calculation with the learned anomalous features. Moreover, local-to-global feature

---

[1]https://github.com/mlfoundations/open_clip
[2]https://github.com/facebookresearch/dinov2

Table 5: **Effect of anomaly-aware local-to-global feature fusion.** The local-to-global feature aggregation yields higher performance suggesting its more effective capture of anomalous cues.

|  | Image-level | |
| --- | --- | --- |
| Approach | **MVTec-AD** | **VisA** |
| Simple Score Aggregation | (92.7, 96.4, 93.2) | (84.1, 87.2, 81.8) |
| Score-based Feature Aggregation | (**93.8**, **97.5**, **93.8**) | (**85.3**, **87.9**, **82.6**) |

aggregation not only amplifies anomalous cues that are already captured by the image-level feature, but also incorporates anomalous regions ignored by the less anomaly sensitive attention of the original CLIP, providing richer and more holistic image-level features.

## C State-of-the-Art Methods

We compare our proposed model with five state-of-the-art methods from the literature. The reported results are either sourced from the respective papers or reproduced if unavailable. An overview of their approaches and reproduction details is as follows:

- **AnoVL Deng et al. (2023)**: Adapts vision-language models for zero-shot anomaly detection by optimizing model parameters through test-time adaptation. It uses v-v attention Li et al. (2023) to address spatial misalignment of textual and patch embeddings. AUROC, PRO, and F1-Max values on MVTec-AD and VisA are from the original paper; other metrics and datasets are derived from the official code implementation.

- **VAND Chen et al. (2023)**: Utilizes a vision encoder fine-tuning strategy with linear projections atop features from an auxiliary training set. Its textual prompting approach is similar to WinCLIP and AnoVL. Metrics for MVTec-AD and VisA are from the original publication. For datasets without official measurements, results are reproduced using default parameters from their paper.

- **AnomalyCLIP Zhou et al. (2024)**: Introduces object-agnostic learnable prompts for zero-shot anomaly detection. By using a general [object] token in text prompts, it emphasizes anomalous regions across domains, enhancing generalization without category-specific training. The original paper provides metrics for all datasets except F1-Max, which are reproduced using the official codebase and default parameters.

- **AdaCLIP Cao et al. (2024)**: Adapts CLIP for zero-shot anomaly detection by incorporating learnable deep tokens into vision and text encoders. It uses static and dynamic prompts; static prompts are shared across images for preliminary adaptation, while dynamic prompts are image-specific. The original paper provides metrics for all datasets except AUPRO and Image-AP, which are reproduced using the official codebase and default parameters.

- **AA-CLIP** Ma et al. (2025): Applies a two-stage adaptation strategy, which first fine-tunes the text encoder via residual adapters while freezing the vision encoder, and then fine-tunes the vision encoder using residual adapters and projectors while freezing the text tower. For datasets (KSDD, DTD-Synthetic, DAGM) and metrics (AUPRO and image-/pixel-level F1-max) that lack official numbers, we reproduce the results using the authors' default settings and official repository.

## D Datasets

For a comprehensive evaluation of the zero-shot generalization capabilities of the proposed model, we conduct experiments across test set of 14 diverse datasets, encompassing two domains (industrial and medical) and three modalities (photography, radiology, and endoscopy). Table 6 provides a detailed listing of the statistical details, application, modality, and anomaly pattern types.As shown, some datasets are solely applicable for

Table 6: **Numerical details of the utilized datasets.** $|\mathcal{C}|$ indicates the number of object categories in each dataset. The "Labels" column specifies whether the dataset contains image-level and/or pixel-level annotations.

| Category | Dataset | Type | Modalities | $|\mathcal{C}|$ | # Normal & Anomalous | Detection | Task detection | localization |
|---|---|---|---|---|---|---|---|---|
| Industrial | MVTec AD | Obj & texture | Photography | 15 | (467, 1258) | Industrial defect | ✓ | ✓ |
| | VisA | Obj | Photography | 12 | (962, 1200) | Industrial defect | ✓ | ✓ |
| | MPDD | | Photography | 6 | (176, 282) | Industrial defect | ✓ | ✓ |
| | BTAD | | Photography | 3 | (451, 290) | Industrial defect | ✓ | ✓ |
| | SDD | | Photography | 1 | (181, 74) | Industrial defect | ✓ | ✓ |
| | DAGM | Texture | Photography | 10 | (6996, 1054) | Industrial defect | ✓ | ✓ |
| | DTD-Synthetic | | Photography | 12 | (357, 947) | Industrial defect | ✓ | ✓ |
| Medical | ISIC | Skin | Photography | 1 | (0, 379) | Skin cancer | ✗ | ✓ |
| | CVC-ClinicDB | Colon | Endoscopy | 1 | (0, 612) | Colon polyp | ✗ | ✓ |
| | CVC-ColonDB | | Endoscopy | 1 | (0, 380) | Colon polyp | ✗ | ✓ |
| | Endo | | Endoscopy | 1 | (0, 200) | Colon polyp | ✓ | ✗ |
| | TN3K | Thyroid | Radiology (Ultrasound) | 1 | (0, 614) | Thyroid nodule | ✗ | ✓ |
| | HeadCT | Brain | Radiology (CT) | 1 | (100, 100) | Brain tumor | ✓ | ✗ |
| | BrainMRI | | Radiology (MRI) | 1 | (98, 155) | Brain tumor | ✓ | ✗ |
| | Br35H | | Radiology (MRI) | 1 | (1500, 1500) | Brain tumor | ✓ | ✗ |

anomaly localization or detection because they either contain only abnormal images with segmentation masks (e.g., CVC-ColonDB) or include both normal and abnormal samples but lack pixel-level masks.

Table 7: **Comparisons of ZSAD methods in the medical domain, Supervised models are trained on medical datasets.** The best performance is **bold**, and the second-best is underlined.

| Metric | Dataset | WinCLIP[†] | AnoVL[†] | VAND | AnomalyCLIP | AdaCLIP | **Crane**[+] |
|---|---|---|---|---|---|---|---|
| Image-level ↑ (AUROC, AP, F1-max) | HeadCT | (81.8, 80.2, 79.8) | (82.3, 81.2, 79.1) | (89.2, 89.5, 82.1) | (93.5, 95.1, 91.7) | (81.5, 85.9 ,75.3) | (**96.8**, **97.5**, **92.0**) |
| | BrainMRI | (86.6, 91.5, 86.3) | (84.3, 89.2, 84.8) | (89.6, 91.0, 88.5) | (95.5, 97.2, 93.5) | (61.5, 73.5 ,76.6) | (**97.8**, **98.7**, **95.3**) |
| | Br35H | (80.5, 82.2, 74.4) | (80.0, 80.7, 75.2) | (91.4, 91.9, 84.2) | (**97.9**, 98.0, 92.5) | (52.4, 58.3 ,67.5) | (97.3, **98.1**, **93.7**) |
| | Average | (83.0, 84.6, 80.2) | (82.2, 83.7, 79.7) | (90.1, 90.8, 84.9) | (95.6, 96.8, 92.6) | (65.1, 72.6, 73.1) | (**97.3**, **98.1**, **93.7**) |
| Pixel-level ↑ (AUROC, AUPRO, F1-max) | ISIC | (83.3, 55.1, 48.5) | (**92.6**, 82.2, 76.6) | (83.1, 70.5, 63.7) | (83.0, 63.8, 66.1) | (72.8, 3.23, 55.0) | (91.5, **84.1**, **77.6**) |
| | ColonDB | (70.3, 32.5, 19.6) | (76.2, 44.1, 26.8) | (88.7, 82.5, 58.8) | (87.5, 78.5, 52.1) | (85.5, 24.1, 56.3) | (**94.2**, **85.3**, **59.3**) |
| | ClinicDB | (51.2, 13.8, 24.4) | (79.7, 51.4, 36.3) | (**93.5**, **86.6**, **71.9**) | (92.4, 82.9, 60.0) | (92.3, 54.1, 69.6) | (91.3, 84.6, 69.1) |
| | TN3K | (70.7, 39.8, 30.0) | (70.2, 34.4, 32.3) | (76.9, 37.2, 40.5) | (79.2, 47.0, 47.6) | (52.4, 0.45, 23.0) | (**87.2**, **54.8**, **49.6**) |
| | Average | (68.9, 35.3, 30.6) | (79.7, 53.0, 43.0) | (85.6, 69.2, 58.7) | (85.5, 68.1, 56.4) | (75.8, 20.5, 51.0) | (**91.0**, **77.2**, **63.9**) |

# E    Ablation Studies on Medical Training

Building on the remarkable performance of the model when applied to medical datasets—despite being trained solely on an industrial dataset with significant texture differences—we examine the model's behavior when medical samples are included in training, while the test domain remains unexposed. Following the approach in Zhou et al. (2024), we combine the test split of the classification EndoTect Hicks et al. (2021) with CVC-ColonDB Tajbakhsh et al. (2015) and evaluate on other medical datasets. For evaluation on CVC-ColonDB, we train the model on CVC-ClinicDB Bernal et al. (2015) and test it on the EndoTect test dataset. Although both CVC-ColonDB and CVC-ClinicDB consist of colonoscopy images, they were captured using different endoscopic equipment, resulting in variations in image quality and texture.

Table 7 presents the performance of the selected models under the aforementioned training scheme. Compared to the default evaluation in Main Table 1, Our full model, Crane[+] achieves notable gains in pixel-level anomaly detection, with improvements of 12.1% in F1-max and 5.6% in AUROC. At the image level, it records increases of 2.3% in F1-max and 1.5% in AUROC. Furthermore, in the current table, Crane[+] surpasses the second-best model by 5.4% in AUROC and 5.2% in F1-max for pixel-level performance, and by 1.7% in AUROC and 1.3% in F1-max at the image level.

# F    Backbones in D-Attn : SAM vs. DINOv2

To further validate the effectiveness of Crane[+] , we evaluated the impact of incorporating SAM Kirillov et al. (2023) visual features in the D-Attn module, as summarized in Table 8. The results reflect mean performance across the seven industrial datasets presented in Table 1. While SAM-based features offer strong baseline performance, DINOv2 features consistently lead to better outcomes across both pixel- and image-level tasks, indicating superior semantic representation for anomaly localization. Additionally, DINOv2 provides higher inference efficiency, reinforcing its suitability as the visual backbone in our pipeline.

Table 8: **DINOv2 vs. SAM performance comparison.** Comparison of SAM and DINOv2 visual features in the D-Attn module of Crane[+] . DINOv2 achieves stronger anomaly localization and classification performance with improved inference efficiency, averaged across seven industrial datasets.

| Model | Pixel-level (ROC, PRO, F1-M) | Image-level (ROC, AP, F1-M) | Infr. (FPS) |
|---|---|---|---|
| SAM-ViT-B | 95.7, 89.4, 52.3 | 91.7, 91.7, 89.9 | 2.65 |
| DINOv2-B14 | **96.4**, **92.3**, **54.0** | **92.5**, **93.5**, **90.1** | **3.38** |

# G Category-level Results

In this section, we report quantitative results for both `Crane` and `Crane`$^+$ across individual sub-dataset categories, alongside the main aggregated results. This breakdown captures the performance distribution across categories, revealing category-specific behaviors and offering a more nuanced view of model robustness. We also include qualitative results for the full model, `Crane`$^+$, to aid interpretation of outlier cases. All results are reproduced according to the implementation details in Appendix A.

## G.1 Category-level Quantitative Results

Table 9: **Category-level anomaly localization performance for the dataset BTAD.** each triplet reports (AUROC, AUPRO, F1-max).

| Product | AnVoL | VAND | AnomalyCLIP | AdaCLIP | Crane | Crane$^+$ |
|---|---|---|---|---|---|---|
| 01 | (93.8, 56.0, 46.2) | (89.9, 72.3, 53.3) | (93.7, 73.0, 52.9) | (88.8, 1.60, 54.1) | (96.2, 77.3, 60.8) | (97.0, 84.0, 65.5) |
| 02 | (57.8, 15.8, 17.1) | (86.3, 50.3, 56.7) | (94.4, 66.0, 60.0) | (95.9, 12.0, 64.0) | (95.7, 72.8, 64.0) | (95.0, 81.1, 66.7) |
| 03 | (74.0, 50.9, 6.90) | (91.8, 83.6, 12.0) | (94.6, 87.1, 36.4) | (96.3, 47.0, 38.4) | (97.9, 93.6, 45.9) | (97.9, 95.5, 51.3) |
| **Mean** | (75.2, 40.9, 23.4) | (89.3, 68.7, 40.6) | (94.2, 75.4, 49.7) | (92.9, 22.8, 48.1) | (96.6, 81.3, 56.9) | (96.7, 86.9, 61.2) |

Table 10: **Category-level anomaly classification performance for the dataset BTAD.** each triplet reports (AUROC, AP, F1-max).

| Product | AnVoL | VAND | AnomalyCLIP | AdaCLIP | Crane | Crane$^+$ |
|---|---|---|---|---|---|---|
| 01 | (94.8, 97.9, 92.8) | (82.0, 92.3, 84.3) | (90.9, 96.6, 89.4) | (93.2, 95.6, 91.8) | (98.3, 99.3, 95.7) | (98.2, 99.3, 96.8) |
| 02 | (65.4, 93.8, 93.0) | (82.0, 96.8, 93.5) | (84.1, 97.4, 93.3) | (80.3, 94.8, 87.2) | (86.6, 97.9, 93.7) | (92.2, 98.8, 94.7) |
| 03 | (80.7, 27.6, 33.2) | (57.5, 19.8, 26.4) | (89.8, 70.7, 68.6) | (97.6, 93.6, 85.6) | (98.3, 90.5, 85.7) | (98.5, 92.6, 87.2) |
| **Mean** | (80.3, 72.8, 73.0) | (73.8, 69.6, 68.1) | (88.2, 87.3, 83.8) | (88.7, 93.8, 88.2) | (94.4, 95.9, 91.7) | (96.3, 96.9, 92.9) |

Table 11: **Category-level anomaly localization performance for the dataset DAGM.** each triplet reports (AUROC, AUPRO, F1-max).

| Product | AnVoL | VAND | AnomalyCLIP | AdaCLIP | **Crane** | **Crane$^+$** |
|---|---|---|---|---|---|---|
| Class01 | (58.0, 18.1, 2.0 ) | (70.5, 52.0, 32.5) | (88.0, 76.7, 50.2) | (83.5, 58.8, 44.8) | (88.3, 72.7, 48.5) | (86.6, 75.7, 40.2) |
| Class02 | (90.0, 75.5, 16.4) | (88.1, 83.5, 51.1) | (99.5, 99.1, 64 ) | (96.2, 66.5, 66.6) | (99.6, 98.2, 73.1) | (98.7, 98.8, 72.5) |
| Class03 | (86.8, 62.2, 7.2 ) | (78.5, 61.5, 36.7) | (95.9, 93.8, 69.7) | (96.2, 44.3, 72.7) | (95.8, 89.9, 73.2) | (95.6, 93.6, 70.5) |
| Class04 | (79.4, 54.7, 8.1 ) | (75.5, 44.1, 5.3 ) | (89.1, 75.3, 35.6) | (86.7, 17.1, 9.6 ) | (92.5, 80.8, 35.6) | (93.9, 86.4, 33.6) |
| Class05 | (81.8, 56.5, 13.5) | (82.3, 64.0, 54.9) | (99.1, 96.9, 74.2) | (97.5, 52.0, 76.7) | (99.1, 94.6, 79.5) | (99.3, 98.1, 80.5) |
| Class06 | (93.1, 81.2, 47.6) | (91.9, 81.7, 74.5) | (99.1, 96.0, 76.4) | (99.1, 61.5, 82.7) | (99.3, 93.2, 78.8) | (99.8, 99.1, 82.8) |
| Class07 | (63.3, 26.9, 3.5 ) | (83.6, 69.6, 54.2) | (90.3, 86.5, 70 ) | (93.3, 57.4, 72.1) | (90.5, 87.8, 72.8) | (89.9, 88.7, 70.9) |
| Class08 | (58.6, 23.0, 0.3 ) | (80.2, 64.1, 12.1) | (98.3, 96.3, 55.7) | (93.5, 40.1, 64.4) | (99.0, 98.4, 68.4) | (99.6, 99.6, 68.7) |
| Class09 | (89.2, 70.1, 3.9 ) | (90.6, 78.6, 26.9) | (98.3, 92.7, 33.5) | (94.9, 60.8, 44.0) | (99.9, 99.2, 76.0) | (99.9, 99.6, 75.2) |
| Class10 | (97.0, 91.6, 25.7) | (83.0, 61.1, 25.5) | (98.5, 97.1, 60.1) | (96.1, 47.5, 61.2) | (98.8, 97.1, 66.5) | (99.2, 98.9, 73.4) |
| **Mean** | (79.7, 56.0, 12.8) | (82.4, 66.0, 37.4) | (95.6, 91.0, 58.9) | (93.7, 50.6, 59.5) | (96.3, 91.2, 67.2) | (96.2, 93.8, 66.8) |

Table 12: **Category-level anomaly classification performance for the dataset DAGM.** each triplet reports (AUROC, AP, F1-max).

| Product | AnVoL | VAND | AnomalyCLIP | AdaCLIP | **Crane** | **Crane$^+$** |
|---|---|---|---|---|---|---|
| Class01 | (58.3, 22.8, 30.8) | (79.9, 37.0, 38.2) | (85.8, 49.8, 52.6) | (88.3, 44.2, 51.5) | (92.8, 78.8, 73.2) | (89.9, 66.8, 65.2) |
| Class02 | (99.4, 98.1, 95.1) | (95.0, 87.0, 81.3) | (100 , 100 , 100 ) | (99.5, 98.5, 95.9) | (100 , 100 , 100 ) | (100 , 100 , 100 ) |
| Class03 | (99.6, 97.9, 93.0) | (99.4, 96.6, 91.4) | (99.9, 99.5, 97 ) | (100 , 100 , 100 ) | (100 , 100 , 100 ) | (100 , 100 , 100 ) |
| Class04 | (89.9, 74.5, 54.3) | (85.2, 66.4, 48.6) | (97.6, 94.1, 85.7) | (95.8, 89.2, 79.4) | (99.4, 98.5, 96.4) | (99.0, 98.1, 93.2) |
| Class05 | (95.7, 87.4, 72.8) | (94.6, 82.2, 65.9) | (99.2, 98.1, 100 ) | (98.9, 94.3, 90.2) | (100 , 100 , 100 ) | (99.9, 99.6, 98.2) |
| Class06 | (99.1, 95.8, 87.3) | (98.5, 92.4, 81.6) | (99.8, 99.4, 100 ) | (99.9, 98.7, 96.1) | (100 , 100 , 100 ) | (100 , 99.9, 99.1) |
| Class07 | (78.8, 54.6, 40.5) | (86.1, 70.9, 58.3) | (95.3, 90.5, 94.8) | (94.7, 82.8, 73.6) | (100 , 100 , 99.7) | (98.3, 96.0, 90.7) |
| Class08 | (81.5, 60.3, 44.9) | (90.3, 78.7, 67.2) | (97.7, 95.2, 87.1) | (96.5, 89.1, 83.1) | (99.8, 99.0, 96.3) | (99.5, 98.8, 96.7) |
| Class09 | (92.4, 83.2, 68.1) | (94.8, 88.0, 75.5) | (99.0, 98.3, 86.1) | (98.7, 96.8, 92.5) | (99.7, 98.3, 93.4) | (99.9, 99.7, 98.4) |
| Class10 | (96.8, 90.5, 80.2) | (92.0, 82.1, 70.7) | (98.9, 97.5, 97.6) | (98.5, 95.6, 89.8) | (100 , 99.8, 99.0) | (99.8, 99.3, 98.0) |
| **Mean** | (89.7, 76.1, 74.7) | (94.4, 83.9, 79.9) | (97.7, 92.4, 90.1) | (96.9, 88.5, 87.7) | (99.2, 97.4, 95.8) | (98.9, 96.1, 94.7) |

Table 13: **Category-level anomaly localization performance for the dataset DTD-Syn.** each triplet reports (AUROC, AUPRO, F1-max).

| Product | AnVoL | VAND | AnomalyCLIP | AdaCLIP | **Crane** | **Crane$^+$** |
|---|---|---|---|---|---|---|
| Woven_001 | (93.0, 75.6, 33.8) | (99.2, 82.6, 77.9) | (99.7, 98.9, 67.2) | (99.9, 87.1, 78.0) | (99.8, 99.1, 72.7) | (99.9, 99.4, 75.0) |
| Woven_127 | (89.4, 74.9, 19.1) | (90.8, 55.6, 60.2) | (93.7, 89.5, 51.9) | (96.0, 65.3, 64.2) | (95.6, 94.7, 63.6) | (95.3, 93.7, 65.9) |
| Woven_104 | (96.1, 86.5, 41.5) | (94.3, 69.5, 68.9) | (96.1, 92.5, 63.1) | (98.6, 79.4, 73.1) | (98.4, 96.7, 68.7) | (98.6, 96.8, 70.9) |
| Stratified_154 | (99.2, 94.6, 61.1) | (96.8, 77.6, 78.6) | (99.5, 96.2, 67.4) | (97.5, 76.5, 72.4) | (99.5, 99.0, 72.3) | (99.3, 98.3, 72.9) |
| Blotchy_099 | (94.4, 84.1, 37.3) | (99.0, 71.0, 68.5) | (99.5, 96.2, 67.5) | (99.7, 87.3, 79.2) | (99.6, 96.5, 73.2) | (99.7, 97.2, 76.0) |
| Woven_068 | (97.2, 89.1, 33.1) | (95.2, 63.4, 62.9) | (98.7, 92.8, 47.8) | (98.4, 65.1, 60.7) | (98.7, 95.7, 51.6) | (99.1, 96.9, 61.4) |
| Woven_125 | (90.4, 80.8, 33.8) | (98.8, 84.6, 83.5) | (99.4, 95.6, 64.1) | (99.8, 90.6, 82.5) | (99.6, 97.9, 70.5) | (99.7, 99.1, 75.5) |
| Marbled_078 | (97.7, 92 , 43.6) | (98.1, 77.4, 73.3) | (99.1, 97.1, 62 ) | (99.6, 85.2, 77.1) | (99.4, 97.7, 68.3) | (99.4, 97.6, 71.3) |
| Perforated_037 | (98.8, 95.9, 46.3) | (89.0, 61.0, 68.1) | (94.6, 85.1, 63.1) | (96.4, 70.6, 69.2) | (96.6, 94.1, 70.0) | (97.9, 96.5, 71.7) |
| Mesh_114 | (83.4, 57.7, 26.4) | (89.0, 60.6, 66.4) | (95.2, 77.0, 56.5) | (97.7, 73.7, 70.0) | (95.2, 83.3, 64.8) | (97.0, 88.7, 68.1) |
| Fibrous_183 | (93.8, 80.2, 35.6) | (97.5, 56.1, 55.7) | (99.4, 98.2, 69.2) | (99.0, 82.1, 75.0) | (99.7, 99.2, 78.6) | (99.7, 99.1, 77.1) |
| Matted_069 | (86.1, 61.6, 17.5) | (95.2, 44.1, 45.1) | (99.6, 84.8, 66.7) | (98.5, 56.4, 55.5) | (99.7, 88.4, 74.5) | (99.7, 89.2, 75.9) |
| **Mean** | (93.3, 81.1, 35.8) | (95.2, 66.9, 67.4) | (97.9, 92.0, 62.2) | (98.4, 76.6, 71.4) | (98.5, 95.2, 69.1) | (98.8, 96.0, 71.8) |

Table 14: **Category-level anomaly classification performance for the dataset DTD-Syn.** each triplet reports (AUROC, AP, F1-max).

| Product | AnVoL | VAND | AnomalyCLIP | AdaCLIP | Crane | Crane⁺ |
|---|---|---|---|---|---|---|
| Woven_001 | (96.7, 94.0, 98.7) | (96.1, 95.5, 98.6) | (100, 100 , 100 ) | (100 , 100 , 100 ) | (99.4, 99.9, 98.7) | (98.4, 99.6, 97.5) |
| Woven_127 | (80.8, 77.9, 83.8) | (74.4, 70.2, 77.8) | (80.7, 83.5, 76.2) | (99.8, 99.9, 98.8) | (100 , 100 , 100 ) | (98.6, 99.7, 98.1) |
| Woven_104 | (94.8, 93.9, 98.7) | (76.2, 89.9, 93.7) | (98.1, 99.6, 97.5) | (97.9, 99.4, 98.2) | (98.6, 99.7, 98.7) | (100 , 100 , 99.3) |
| Stratified_154 | (99.8, 98.8, 99.9) | (97.4, 96.3, 99.4) | (97.6, 99.4, 95.8) | (91.7, 97.7, 95.1) | (99.7, 99.9, 99.8) | (99.8, 100 , 99.4) |
| Blotchy_099 | (99.3, 98.7, 99.8) | (92.6, 92.0, 98.2) | (98.9, 99.7, 98.8) | (89.6, 95.4, 90.5) | (92.6, 94.3, 87.4) | (90.5, 92.6, 83.3) |
| Woven_068 | (86.2, 81.7, 92.6) | (84.5, 80.0, 91.6) | (96.9, 98.4, 94.9) | (92.6, 98.2, 93.1) | (99.0, 99.8, 98.8) | (100 , 100 , 100 ) |
| Woven_125 | (99.7, 99.4, 99.9) | (94.3, 93.9, 98.5) | (99.8, 100 , 99.4) | (99.4, 99.9, 98.1) | (100 , 100 , 100 ) | (98.9, 99.8, 98.8) |
| Marbled_078 | (98.7, 98.1, 99.7) | (98.8, 98.1, 99.7) | (98.7, 99.7, 97.5) | (99.1, 99.6, 97.8) | (93.7, 98.5, 93.3) | (95.2, 98.9, 94.5) |
| Perforated_037 | (99.9, 99.4, 100 ) | (75.1, 88.9, 92.9) | (90.6, 97.5, 92.5) | (89.5, 94.0, 84.7) | (87.6, 95.1, 86.6) | (88.3, 95.2, 85.9) |
| Mesh_114 | (88.2, 86.1, 95.4) | (72.7, 81.7, 87.7) | (85.8, 94.5, 84.4) | (99.3, 99.8, 99.4) | (95.1, 97.3, 91.0) | (94.8, 97.2, 90.5) |
| Fibrous_183 | (98.1, 97.5, 99.6) | (89.4, 92.8, 97.2) | (97.2, 99.3, 95.7) | (100 , 100 , 100 ) | (99.1, 99.8, 97.5) | (97.3, 99.4, 96.2) |
| Matted_069 | (96.6, 94.6, 99.2) | (74.7, 88.8, 92.5) | (82.6, 95.2, 91.2) | (85.1, 83.5, 81.8) | (91.1, 97.8, 92.2) | (87.8, 96.8, 91.8) |
| **Mean** | (94.9, 93.3, 97.3) | (85.5, 89.0, 94.0) | (93.9, 97.2, 93.6) | (95.3, 97.3, 94.8) | (96.3, 98.5, 95.3) | (95.8, 98.3, 94.6) |

Table 15: **Category-level anomaly localization performance for the dataset MPDD.** each triplet reports (AUROC, AUPRO, F1-max).

| Product | AnVoL | VAND | AnomalyCLIP | AdaCLIP | Crane | Crane⁺ |
|---|---|---|---|---|---|---|
| bracket_black | (24.6, 1.6 , 0.2 ) | (96.3, 90.6, 15.8) | (95.7, 85.2, 27.2) | (96.5, 82.6, 14.4) | (96.1, 86.1, 28.6) | (97.2, 90.4, 30.4) |
| bracket_brown | (27.4, 3.4 , 1.0 ) | (87.4, 72.6, 8.7 ) | (94.4, 77.8, 13.1) | (93.3, 28.7, 18.8) | (99.8, 96.2, 35.1) | (99.8, 97.7, 31.4) |
| bracket_white | (45.2, 1.7 , 0.1 ) | (99.2, 93.7, 8.9 ) | (99.8, 98.8, 22.9) | (98.1, 63.9, 6.9 ) | (97.7, 92.2, 25.2) | (98.0, 95.5, 36.6) |
| connector | (90.0, 68.2, 4.7 ) | (90.6, 74.5, 22.5) | (97.2, 89.9, 27.0) | (97.4, 77.9, 39.2) | (95.6, 82.1, 17.2) | (96.1, 89.1, 16.2) |
| metal_plate | (95.9, 85.3, 70.4) | (93.0, 74.5, 63.1) | (93.7, 86.8, 61.9) | (92.0, 33.5, 57.8) | (93.9, 83.4, 62.5) | (95.3, 89.3, 66.8) |
| tubes | (90.7, 69.7, 17.3) | (99.1, 96.9, 68.7) | (98.1, 93.6, 53.3) | (98.9, 90.4, 70.0) | (98.9, 95.7, 61.0) | (99.2, 97.1, 70.2) |
| **Mean** | (62.3, 38.3, 15.6) | (94.3, 83.8, 31.3) | (96.5, 88.7, 34.2) | (96.0, 62.8, 34.5) | (97.0, 89.3, 38.2) | (97.6, 93.2, 41.9) |

Table 16: **Category-level anomaly classifcation performance for the dataset MPDD.** each triplet reports (AUROC, AP, F1-max).

| Product | AnVoL | VAND | AnomalyCLIP | AdaCLIP | Crane | Crane⁺ |
|---|---|---|---|---|---|---|
| bracket_black | (42.5, 59.7, 76.5) | (68.4, 72.6, 80.0) | (67.8, 73.4, 78.6) | (71.4, 81.1, 77.7) | (82.1, 74.1, 66.7) | (83.3, 70.3, 71.8) |
| bracket_brown | (66.7, 92.3, 90.7) | (61.6, 78.0, 81.0) | (62.0, 80.4, 80.3) | (51.9, 71.8, 79.7) | (80.8, 81.8, 76.9) | (84.0, 82.7, 82.5) |
| bracket_white | (38.5, 58.2, 76.5) | (85.7, 88.2, 78.1) | (67.7, 71.6, 69.8) | (77.8, 80.0, 74.7) | (72.9, 77.7, 81.1) | (76.9, 81.8, 79.6) |
| connector | (100 , 100 , 100 ) | (78.5, 71.7, 66.7) | (87.4, 77.0, 73.7) | (64.4, 61.9, 58.3) | (64.9, 80.1, 81.0) | (54.6, 74.4, 79.7) |
| metal_plate | (98.3, 99.4, 95.9) | (69.9, 86.5, 86.6) | (84.7, 94.4, 87.5) | (86.6, 94.8, 90.3) | (91.4, 96.8, 90.9) | (89.9, 96.6, 89.7) |
| tubes | (90.4, 95.4, 90.4) | (95.7, 98.1, 92.2) | (95.4, 98.1, 92.3) | (80.8, 91.6, 84.7) | (96.4, 98.5, 93.0) | (97.5, 99.0, 94.8) |
| **Mean** | (72.7, 83.6, 88.3) | (76.6, 82.5, 80.8) | (77.5, 82.5, 80.4) | (72.1, 80.2, 77.6) | (81.4, 84.8, 81.6) | (81.0, 84.1, 83.0) |

Table 17: **Category-level anomaly localization performance for the dataset MVTec-AD.** each triplet reports (AUROC, AUPRO, F1-max).

| Product | AnVoL | VAND | AnomalyCLIP | AdaCLIP | Crane | Crane$^+$ |
|---|---|---|---|---|---|---|
| bottle | (90.9, 75.7, 53.2) | (83.5, 45.6, 53.4) | (90.4, 80.8, 51.6) | (90.8, 57.6, 60.8) | (91.6, 84.6, 54.6) | (93.6, 88.2, 60.6) |
| grid | (67.7, 66.3, 23.0) | (72.3, 25.7, 23.9) | (78.9, 64.0, 18.9) | (78.3, 35.3, 26.5) | (97.7, 80.0, 35.7) | (99.2, 95.5, 48.1) |
| carpet | (82.8, 52.4, 14.9) | (92.0, 51.3, 33.1) | (95.8, 87.6, 31.0) | (95.2, 18.0, 32.9) | (99.0, 91.9, 59.2) | (99.3, 95.8, 66.8) |
| capsule | (95.6, 85.1, 42.5) | (98.4, 48.5, 65.7) | (98.8, 90.0, 57.0) | (98.9, 36.1, 67.4) | (96.0, 90.5, 33.8) | (95.5, 92.4, 33.5) |
| cable | (95.7, 87.1, 22.3) | (95.8, 31.6, 40.8) | (97.3, 75.4, 32.0) | (97.0, 20.3, 39.0) | (77.7, 68.8, 23.1) | (79.4, 71.5, 22.0) |
| hazelnut | (93.8, 75.6, 32.5) | (96.1, 70.2, 50.5) | (97.2, 92.5, 47.6) | (96.5, 59.2, 40.1) | (97.1, 94.3, 50.6) | (97.4, 94.8, 53.2) |
| leather | (98.7, 94.4, 36.3) | (99.1, 72.4, 50.0) | (98.6, 92.2, 33.2) | (99.3, 76.9, 47.7) | (99.1, 97.2, 40.7) | (99.1, 98.3, 41.4) |
| metal_nut | (71.4, 46.5, 29.6) | (65.5, 38.4, 28.0) | (74.6, 71.1, 33.1) | (74.4, 62.4, 35.3) | (72.7, 74.4, 33.4) | (76.9, 80.1, 34.0) |
| screw | (79.5, 69.8, 18.4) | (76.2, 65.4, 27.7) | (91.8, 88.1, 35.5) | (87.7, 27.9, 35.7) | (98.4, 91.7, 33.8) | (98.6, 92.8, 39.3) |
| pill | (88.5, 60.1, 13.5) | (97.8, 67.1, 41.7) | (97.5, 88.0, 33.4) | (98.3, 70.3, 34.5) | (89.1, 90.8, 28.7) | (85.1, 91.3, 27.5) |
| toothbrush | (77.0, 54.3, 35.9) | (92.7, 26.7, 66.5) | (94.7, 87.4, 64.9) | (91.1, 30.1, 61.9) | (93.5, 90.4, 35.8) | (89.7, 90.8, 29.5) |
| wood | (91.6, 80.1, 20.6) | (95.8, 54.5, 48.1) | (91.9, 88.5, 29.0) | (94.7, 69.4, 37.9) | (97.5, 93.8, 60.7) | (96.7, 96.0, 60.8) |
| transistor | (75.6, 50.9, 24.6) | (62.4, 21.3, 19.0) | (70.8, 58.2, 18.8) | (57.8, 31.2, 16.3) | (69.5, 56.0, 17.7) | (65.4, 54.8, 16.7) |
| tile | (95.1, 75.1, 49.7) | (95.8, 31.1, 60.3) | (96.4, 91.5, 55.2) | (92.6, 48.1, 56.0) | (96.1, 87.5, 65.7) | (95.5, 90.3, 68.0) |
| zipper | (94.3, 82.2, 35.1) | (91.1, 10.7, 40.5) | (91.3, 65.4, 45.0) | (95.8, 18.2, 57.2) | (95.1, 77.3, 46.3) | (97.3, 89.3, 54.9) |
| **Mean** | (86.6, 70.4, 30.1) | (87.6, 44.0, 43.3) | (91.1, 81.4, 39.1) | (89.9, 44.1, 43.4) | (91.3, 84.6, 41.3) | (91.2, 88.1, 43.8) |

Table 18: **Category-level anomaly classification performance for the dataset MVTec-AD.** each triplet reports (AUROC, AP, F1-max).

| Product | AnVoL | VAND | AnomalyCLIP | AdaCLIP | Crane | Crane$^+$ |
|---|---|---|---|---|---|---|
| bottle | (96.3, 99.0, 96.8) | (91.8, 97.6, 92.1) | (88.7, 96.8, 90.9) | (97.8, 99.3, 95.4) | (91.3, 97.5, 91.1) | (92.4, 97.7, 92.8) |
| grid | (87.8, 92.0, 86.0) | (88.3, 93.0, 85.1) | (70.3, 81.7, 77.4) | (64.3, 79.2, 76.0) | (99.7, 99.9, 98.3) | (100 , 100 , 100 ) |
| carpet | (79.1, 94.5, 92.7) | (79.8, 95.4, 92.0) | (89.5, 97.8, 91.7) | (84.6, 96.6, 92.0) | (100 , 100 , 100 ) | (99.9, 100 , 99.4) |
| capsule | (99.3, 99.8, 99.4) | (99.5, 99.8, 98.3) | (100 , 100 , 99.4) | (100 , 100 , 100 ) | (93.5, 98.6, 94.1) | (92.3, 98.3, 93.8) |
| cable | (98.1, 99.4, 96.4) | (86.4, 95.0, 89.1) | (97.8, 99.3, 97.3) | (97.7, 99.1, 97.4) | (87.1, 92.5, 83.9) | (88.1, 93.4, 86.2) |
| hazelnut | (94.0, 96.9, 89.4) | (89.5, 94.7, 87.0) | (97.2, 98.5, 92.6) | (87.0, 93.0, 86.1) | (98.5, 99.3, 96.4) | (98.0, 99.1, 96.3) |
| leather | (100 , 100 , 100 ) | (99.7, 99.9, 98.9) | (99.8, 99.9, 99.5) | (99.9, 99.9, 99.5) | (100 , 100 , 100 ) | (100 , 100 , 100 ) |
| metal_nut | (97.4, 99.4, 96.7) | (68.5, 91.9, 89.4) | (92.4, 98.2, 93.7) | (66.6, 92.1, 89.4) | (79.9, 95.5, 90.3) | (84.1, 96.4, 89.9) |
| screw | (86.6, 97.2, 91.8) | (80.6, 96.0, 91.6) | (81.1, 95.3, 92.1) | (88.9, 97.6, 94.0) | (90.0, 95.8, 91.6) | (92.4, 97.3, 92.1) |
| pill | (78.0, 92.1, 87.1) | (84.7, 93.5, 88.8) | (82.1, 92.9, 88.3) | (88.1, 95.0, 90.0) | (84.8, 96.7, 93.5) | (87.0, 97.5, 92.2) |
| toothbrush | (100 , 100 , 99.4) | (99.9, 99.9, 98.8) | (100 , 100 , 100 ) | (100 , 100 , 100 ) | (95.8, 98.7, 95.1) | (86.4, 95.1, 88.5) |
| wood | (92.2, 96.9, 93.1) | (54.0, 72.2, 83.3) | (85.3, 93.9, 90.0) | (91.1, 97.0, 90.9) | (98.6, 99.6, 97.4) | (97.9, 99.4, 97.5) |
| transistor | (86.4, 86.5, 77.9) | (81.1, 77.6, 74.5) | (93.9, 92.1, 83.7) | (86.8, 87.4, 78.7) | (90.6, 88.6, 79.1) | (91.8, 90.2, 79.5) |
| tile | (99.3, 99.8, 98.3) | (99.0, 99.7, 96.8) | (96.9, 99.2, 96.6) | (99.0, 99.7, 96.7) | (99.7, 99.9, 98.8) | (99.7, 99.9, 98.8) |
| zipper | (92.9, 97.9, 93.3) | (89.5, 97.1, 90.8) | (98.4, 99.5, 97.9) | (99.3, 99.8, 98.3) | (97.6, 99.3, 97.9) | (99.0, 99.7, 97.5) |
| **Mean** | (92.5, 96.7, 93.2) | (86.2, 93.6, 90.4) | (91.6, 96.4, 92.7) | (90.1, 95.7, 92.3) | (93.8, 97.5, 93.8) | (93.9, 97.6, 93.6) |

Table 19: **Category-level anomaly localization performance for the dataset ViSA.** each triplet reports (AUROC, AUPRO, F1-max).

| Product | AnVoL | VAND | AnomalyCLIP | AdaCLIP | Crane | Crane$^+$ |
|---|---|---|---|---|---|---|
| candle | (95.6, 83.4, 14.7) | (97.8, 92.5, 39.4) | (98.8, 96.5, 75.6) | (99.2, 76.7, 48.2) | (86.2, 92.2, 16.7) | (90.4, 94.9, 13.5) |
| capsules | (82.9, 44.4, 9.8 ) | (97.5, 86.7, 48.5) | (95.0, 78.9, 82.2) | (98.7, 76.1, 47.6) | (99.0, 87.3, 67.1) | (95.7, 89.3, 32.6) |
| cashew | (89.8, 85.7, 11.1) | (86.0, 91.7, 22.9) | (93.8, 91.9, 80.3) | (95.9, 49.3, 34.2) | (96.8, 88.4, 43.4) | (99.4, 93.7, 69.6) |
| chewinggum | (91.4, 56.0, 49.3) | (99.5, 87.2, 78.5) | (99.3, 90.9, 94.8) | (99.5, 64.9, 79.5) | (94.8, 84.7, 33.7) | (93.8, 90.7, 27.3) |
| fryum | (77.9, 56.4, 23.8) | (92.0, 89.7, 29.7) | (94.6, 86.9, 90.1) | (94.1, 69.2, 28.9) | (98.5, 92.3, 37.6) | (98.9, 97.5, 40.3) |
| macaroni1 | (81.9, 41.0, 1.1 ) | (98.8, 93.2, 35.5) | (98.3, 89.8, 80.4) | (99.7, 82.8, 35.0) | (92.0, 86.8, 15.7) | (93.3, 89.5, 17.5) |
| macaroni2 | (78.0, 34.4, 0.1 ) | (97.8, 82.3, 13.7) | (97.6, 84.0, 71.2) | (99.0, 72.1, 14.2) | (93.7, 81.2, 22.3) | (93.9, 83.7, 25.0) |
| pcb1 | (91.1, 72.1, 17.9) | (92.7, 87.5, 12.5) | (94.0, 80.7, 78.8) | (92.5, 59.0, 23.8) | (89.8, 79.5, 15.6) | (88.4, 79.6, 16.1) |
| pcb2 | (85.1, 54.4, 3.0 ) | (89.7, 75.5, 23.4) | (92.4, 78.9, 67.8) | (92.3, 78.8, 30.6) | (98.9, 90.0, 29.2) | (98.0, 86.0, 7.4 ) |
| pcb3 | (76.0, 32.8, 1.2 ) | (88.4, 77.8, 21.7) | (88.4, 76.8, 66.4) | (87.9, 77.3, 33.7) | (98.3, 86.0, 07.3) | (99.2, 93.0, 34.2) |
| pcb4 | (93.3, 78.4, 33.0) | (94.6, 86.8, 31.3) | (95.7, 89.4, 87.8) | (96.3, 87.5, 43.6) | (98.0, 91.5, 43.2) | (97.4, 97.4, 37.8) |
| pipe_fryum | (79.2, 87.5, 10.8) | (96.0, 90.9, 30.4) | (98.2, 96.2, 89.8) | (97.4, 81.7, 36.0) | (94.9, 90.0, 39.2) | (95.2, 91.6, 40.9) |
| **Mean** | (85.2, 60.5, 14.6) | (94.2, 86.8, 32.3) | (95.5, 86.7, 28.3) | (96.0, 72.9, 37.9) | (95.1, 87.5, 30.9) | (95.3, 90.6, 30.2) |

Table 20: **Category-level anomaly classification performance for the dataset ViSA.** each triplet reports (AUROC, AP, F1-max).

| Product | AnVoL | VAND | AnomalyCLIP | AdaCLIP | Crane | Crane$^+$ |
|---------|-------|------|-------------|---------|-------|-----------|
| candle | (97.2, 97.2, 92.5) | (83.5, 86.6, 77.1) | (80.9, 82.6, 37.8) | (94.5, 95.8, 89.1) | (83.3, 92.9, 82.2) | (85.9, 93.7, 84.5) |
| capsules | (80.6, 89.4, 80.5) | (61.4, 74.5, 78.0) | (82.8, 89.4, 37.8) | (74.3, 82.1, 80.9) | (97.5, 99.0, 96.9) | (81.5, 90.0, 80.2) |
| cashew | (90.2, 95.7, 86.8) | (86.9, 94.0, 84.8) | (76.0, 89.3, 25.8) | (93.7, 97.3, 92.2) | (84.4, 91.2, 83.7) | (97.9, 99.2, 96.4) |
| chewinggum | (96.7, 98.6, 94.8) | (96.5, 98.4, 93.7) | (97.2, 98.8, 61.0) | (91.6, 96.5, 89.6) | (95.1, 97.8, 92.3) | (92.7, 96.8, 90.2) |
| fryum | (90.2, 95.6, 88.7) | (94.2, 97.2, 91.7) | (92.7, 96.6, 30.3) | (86.1, 93.5, 84.7) | (87.1, 89.9, 80.0) | (83.5, 87.2, 76.4) |
| macaroni1 | (70.4, 70.8, 71.4) | (71.4, 70.4, 71.7) | (86.7, 85.5, 23.7) | (73.2, 66.1, 75.7) | (83.6, 84.9, 79.3) | (83.5, 85.5, 77.0) |
| macaroni2 | (61.7, 61.2, 68.9) | (64.7, 63.3, 69.1) | (72.2, 70.8, 5.1 ) | (53.9, 52.8, 68.3) | (74.0, 75.0, 72.5) | (74.7, 76.5, 72.5) |
| pcb1 | (79.7, 82.4, 75.6) | (53.8, 57.4, 66.9) | (85.2, 86.7, 12.7) | (59.7, 64.1, 67.1) | (68.2, 73.0, 66.4) | (57.1, 65.1, 66.4) |
| pcb2 | (56.2, 53.9, 68.1) | (71.6, 73.8, 70.0) | (62.0, 64.4, 15.8) | (53.1, 56.6, 66.7) | (85.4, 87.1, 77.6) | (68.5, 66.4, 69.9) |
| pcb3 | (66.4, 68.2, 69.0) | (66.9, 70.8, 66.7) | (61.7, 69.4, 9.3 ) | (65.4, 69.8, 66.9) | (69.1, 68.3, 71.0) | (85.7, 86.5, 77.2) |
| pcb4 | (75.9, 75.7, 73.6) | (95.1, 95.2, 87.3) | (93.9, 94.3, 34.7) | (77.4, 79.9, 74.6) | (98.7, 99.4, 96.0) | (97.0, 98.5, 94.0) |
| pipe_fryum | (85.4, 92.9, 85.5) | (89.7, 94.7, 87.8) | (92.3, 96.3, 45.5) | (85.9, 92.8, 86.1) | (97.0, 96.8, 93.1) | (94.9, 94.5, 89.2) |
| **Mean** | (79.2, 81.6, 79.6) | (78.0, 81.4, 78.7) | (82.0, 85.3, 80.4) | (75.7, 79.0, 78.5) | (85.3, 87.9, 82.6) | (83.6, 86.6, 81.2) |

## G.2  Category-level Qualitative Results

To provide visual intuition of the method's capability in capturing anomalous patterns, we present zero-shot anomaly maps of $\texttt{Crane}^{+}$ across diverse domains, objects, and textures. For MVTec-AD, we visualize results for the products capsule, carpet, grid, hazelnut, toothbrush, tile, and zipper. For ViSA, we illustrate anomaly maps for the categories candles, capsules, cashew, and pipe_fryum. Localization predictions for the anomalous classes white_brackets, tubes, and plates are reported for MPDD. In DTD-Synthetic, we provide visualizations for the Matted and Fibrous classes, while for DAGM, we include class06, class07, class08, and class09. Additionally, results for the single class from the KSDD dataset are presented. For medical anomaly detection, we provide zero-shot localization outputs for BrainMRI, ISIC, and CVC-ColonDB, using the medical training scheme discussed earlier at Appendix E.

*Limitations.* Despite $\texttt{Crane}$'s strong performance and generalization capabilities, a performance gap remains compared to unsupervised methods that assume access to normal training samples and leverage full-shot training pipelines. This gap partly stems from the inherent challenge zero-shot models face in detecting unseen semantic anomalies, where domain knowledge is crucial to distinguish normal from anomalous samples. Figure 17 illustrates this challenge: for instance, the blue tube is not flagged as anomalous because it lacks general structural defects, yet it is semantically inconsistent within the local context of the dataset.

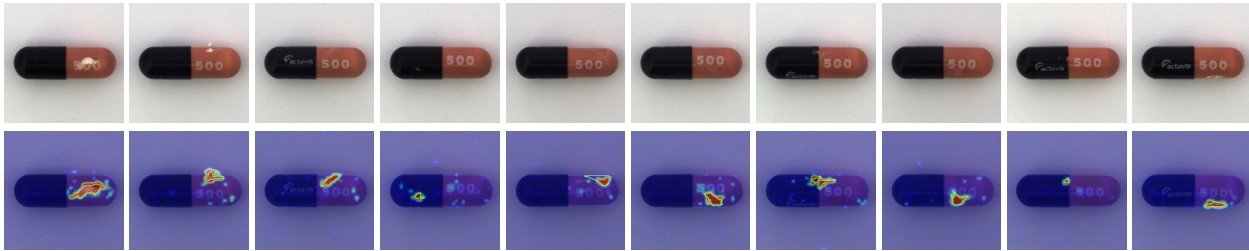

Figure 4: Localization score maps for the product, capsule, in MVTec-AD. The first row illustrates the original image, while the second row shows the anomaly segmentation results, with the regions encircled in green representing the ground truth.

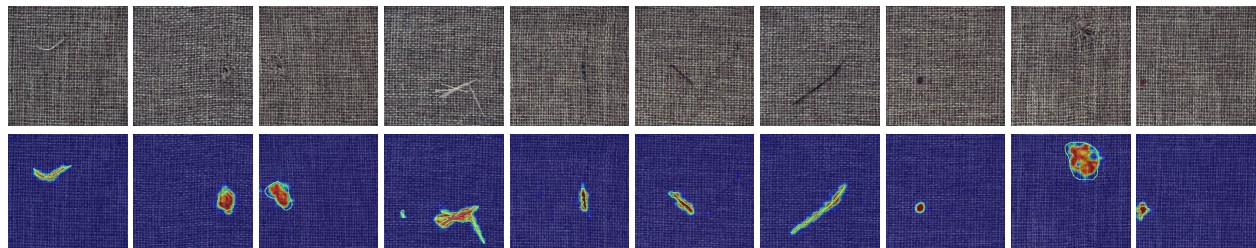

Figure 5: Localization score maps for the product, carpet, in MVTec-AD. The first row illustrates the original image, while the second row shows the anomaly segmentation results, with the regions encircled in green representing the ground truth.

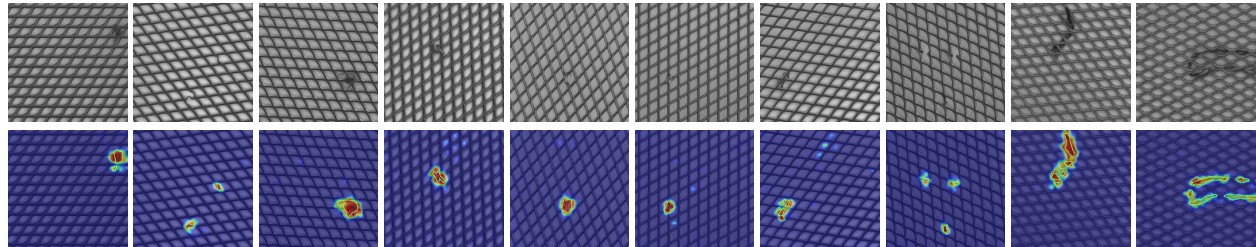

Figure 6: Localization score maps for the product, grid, in MVTec-AD. The first row illustrates the original image, while the second row shows the anomaly segmentation results, with the regions encircled in green representing the ground truth.

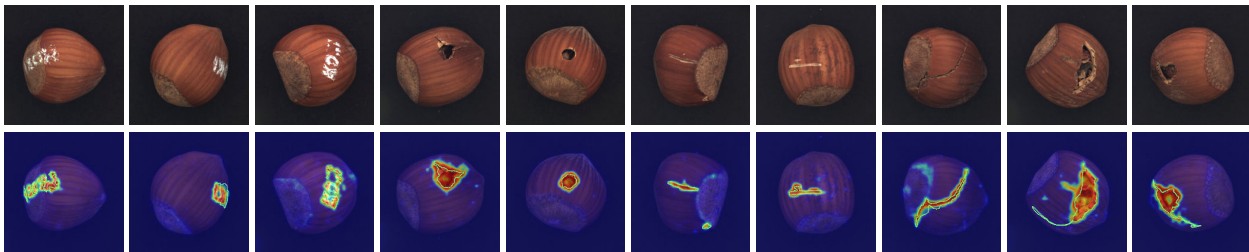

Figure 7: Localization score maps for the product, hazelnut, in MVTec-AD. The first row illustrates the original image, while the second row shows the anomaly segmentation results, with the regions encircled in green representing the ground truth.

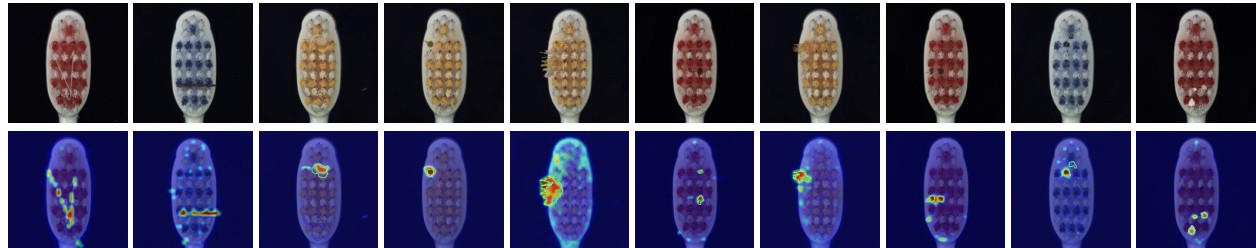

Figure 8: Localization score maps for the product, toothbrush, in MVTec-AD. The first row illustrates the original image, while the second row shows the anomaly segmentation results, with the regions encircled in green representing the ground truth.

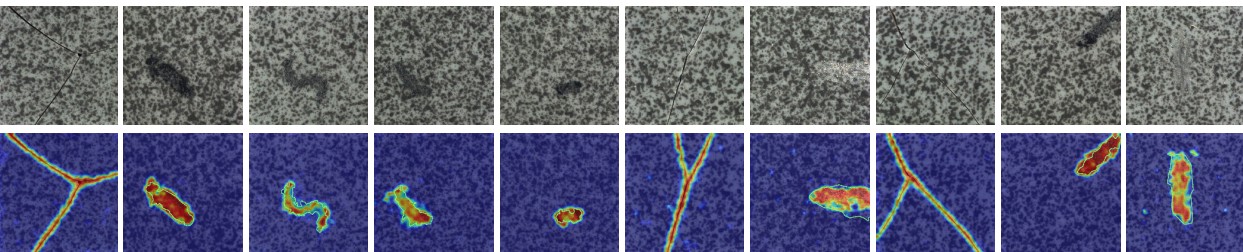

Figure 9: Localization score maps for the product, tile, in MVTec-AD. The first row illustrates the original image, while the second row shows the anomaly segmentation results, with the regions encircled in green representing the ground truth.

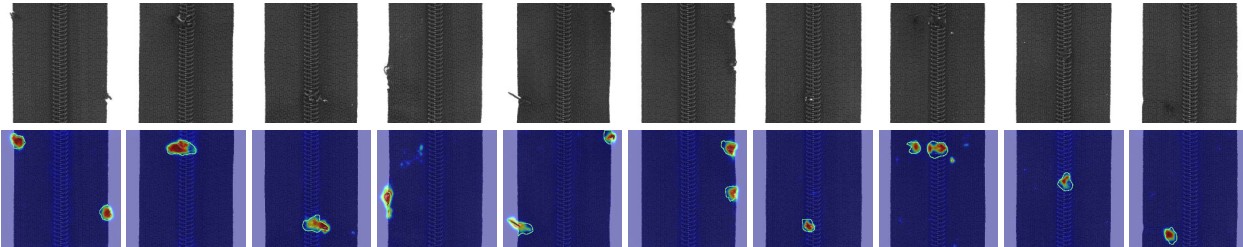

Figure 10: Localization score maps for the product, zipper, in MVTec-AD. The first row illustrates the original image, while the second row shows the anomaly segmentation results, with the regions encircled in green representing the ground truth.

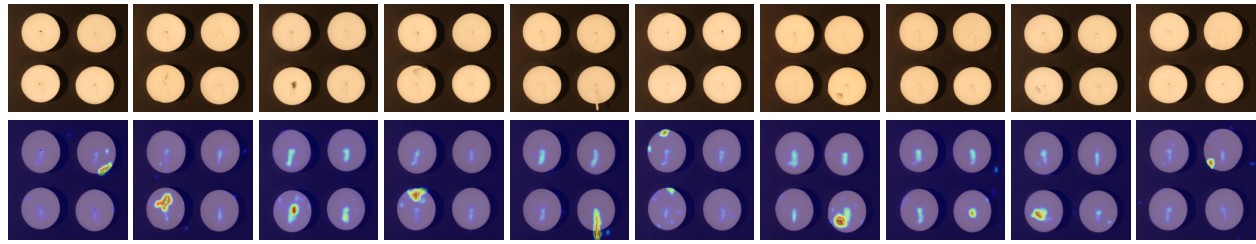

Figure 11: Localization score maps for the product, candles, in VISA dataset. The first row illustrates the original image, while the second row shows the anomaly segmentation results, with the regions encircled in green representing the ground truth.

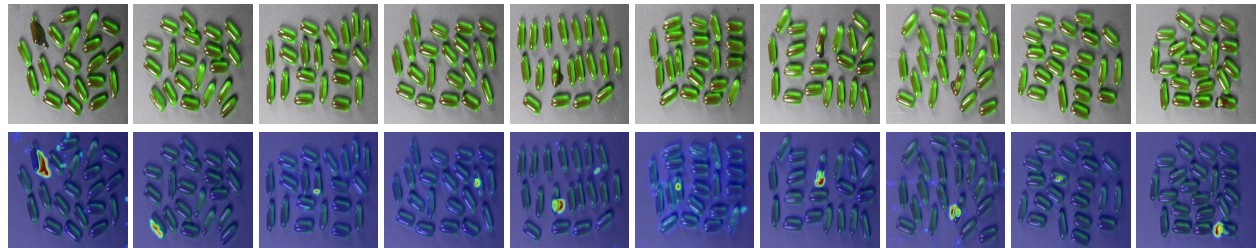

Figure 12: Localization score maps for the product, capsules, in VISA dataset. The first row illustrates the original image, while the second row shows the anomaly segmentation results, with the regions encircled in green representing the ground truth.

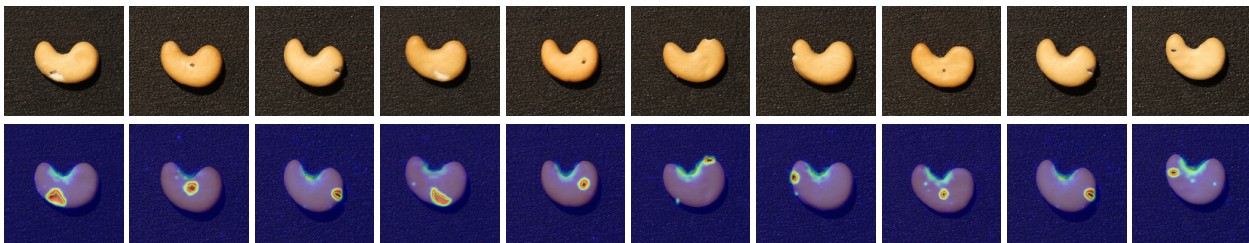

Figure 13: Localization score maps for the product, cashew, in VISA dataset. The first row illustrates the original image, while the second row shows the anomaly segmentation results, with the regions encircled in green representing the ground truth.

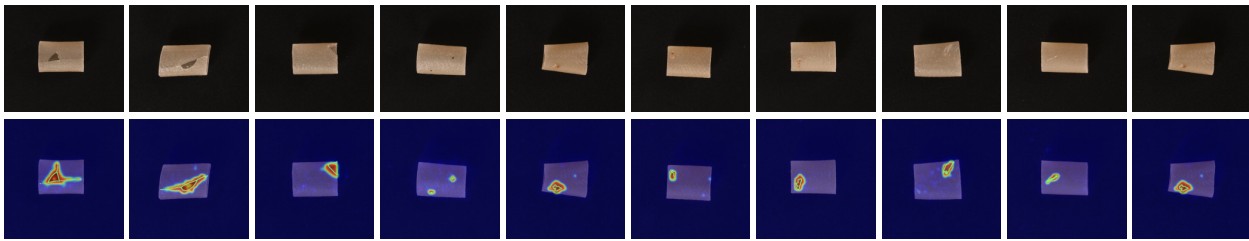

Figure 14: Localization score maps for the product, pipe fryum, in VISA dataset. The first row illustrates the original image, while the second row shows the anomaly segmentation results, with the regions encircled in green representing the ground truth.

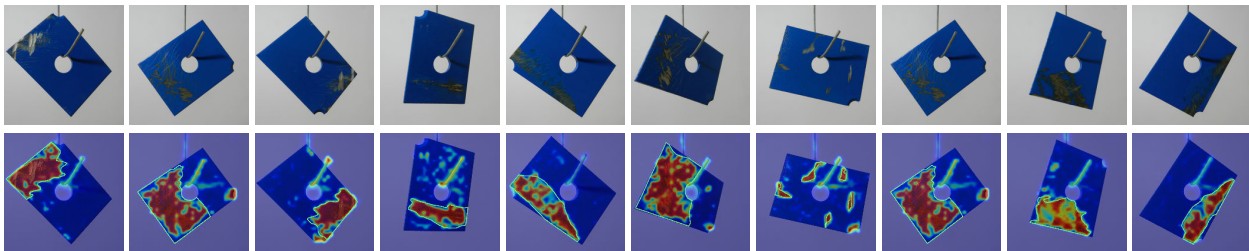

Figure 15: Localization score maps for the product, plates, in MPDD dataset. The first row illustrates the original image, while the second row shows the anomaly segmentation results, with the regions encircled in green representing the ground truth.



Figure 16: Localization score maps for the product, white brackets, in MPDD dataset. The first row illustrates the original image, while the second row shows the anomaly segmentation results, with the regions encircled in green representing the ground truth.

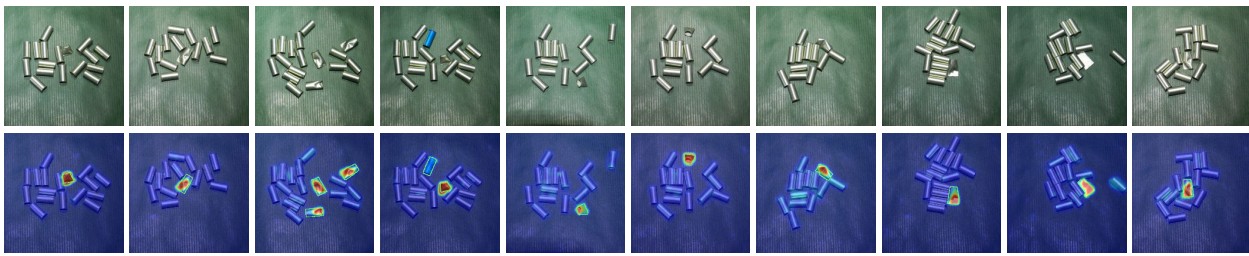

Figure 17: Localization score maps for the product, tubes, in MPDD dataset. The first row illustrates the original image, while the second row shows the anomaly segmentation results, with the regions encircled in green representing the ground truth.

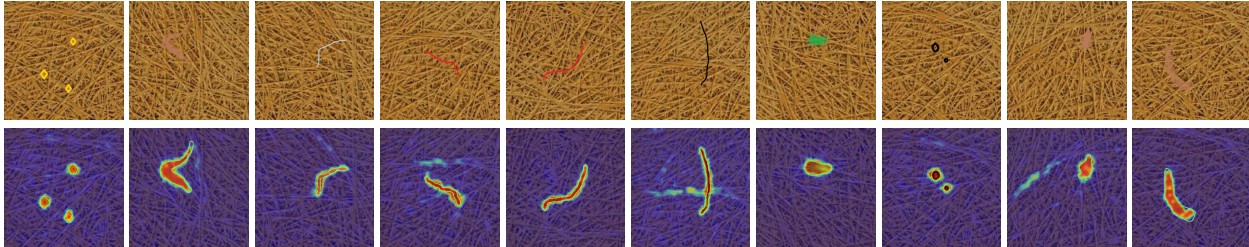

Figure 18: Localization score maps for the product, fibrous, in DTD dataset. The first row illustrates the original image, while the second row shows the anomaly segmentation results, with the regions encircled in green representing the ground truth.

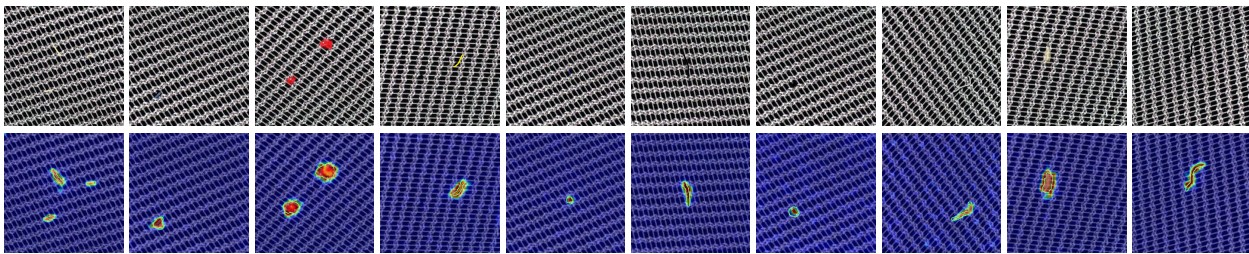

Figure 19: Localization score maps for the product, matted, in DTD dataset. The first row illustrates the original image, while the second row shows the anomaly segmentation results, with the regions encircled in green representing the ground truth.

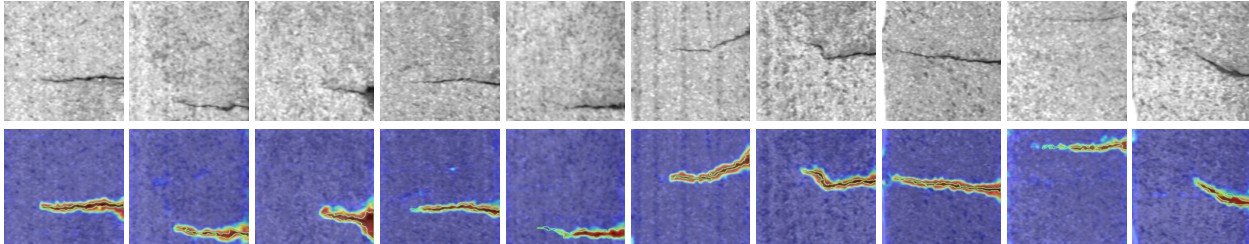

Figure 20: Localization score maps for the product, electric commutators, in SDD dataset. The first row illustrates the original image, while the second row shows the anomaly segmentation results, with the regions encircled in green representing the ground truth.

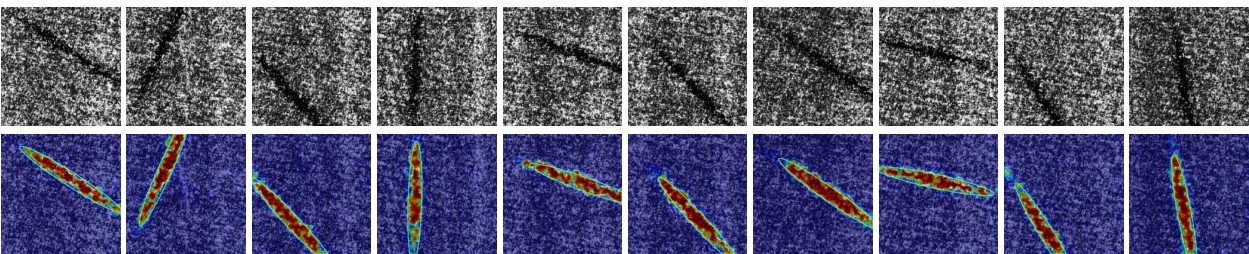

Figure 21: Localization score maps for the product, class 06, in DAGM dataset. The first row illustrates the original image, while the second row shows the anomaly segmentation results, with the regions encircled in green representing the ground truth.

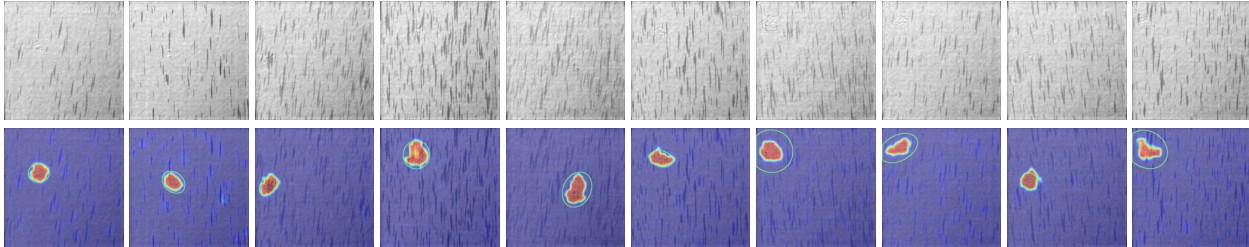

Figure 22: Localization score maps for the product, class 7, in DAGM dataset. The first row illustrates the original image, while the second row shows the anomaly segmentation results, with the regions encircled in green representing the ground truth.

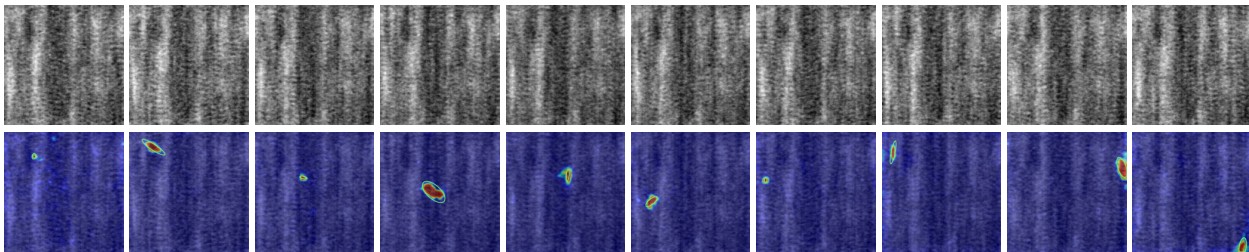

Figure 23: Localization score maps for the product, class 08, in DAGM dataset. The first row illustrates the original image, while the second row shows the anomaly segmentation results, with the regions encircled in green representing the ground truth.

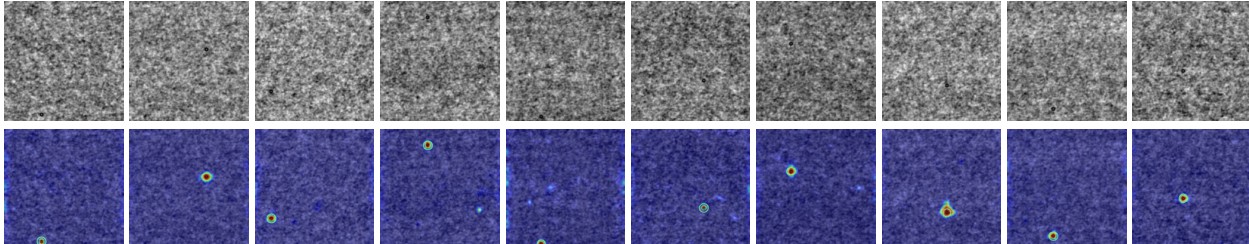

Figure 24: Localization score maps for the product, class 09, in DAGM dataset. The first row illustrates the original image, while the second row shows the anomaly segmentation results, with the regions encircled in green representing the ground truth.

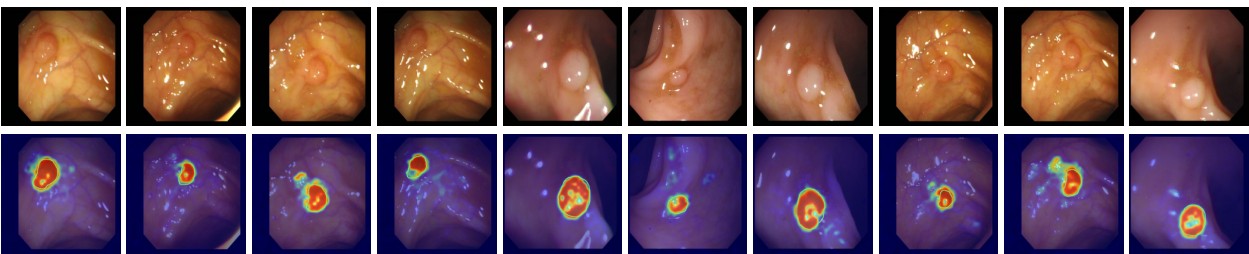

Figure 25: Localization score maps for the ColonDB dataset. The first row illustrates the original image, while the second row shows the anomaly segmentation results, with the regions encircled in green representing the ground truth.

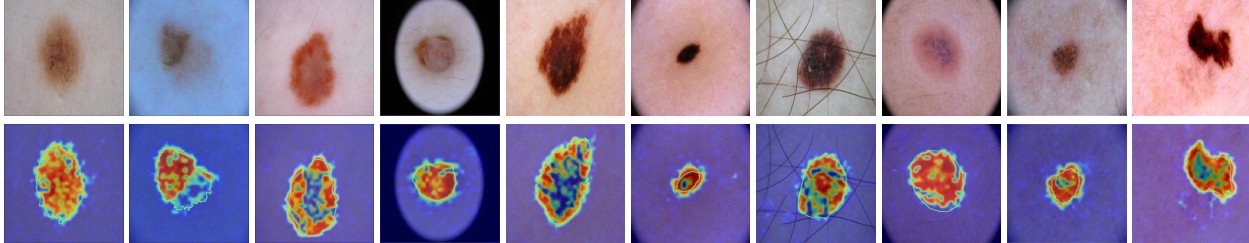

Figure 26: Localization score maps for the ICIC dataset. The first row illustrates the original image, while the second row shows the anomaly segmentation results, with the regions encircled in green representing the ground truth.

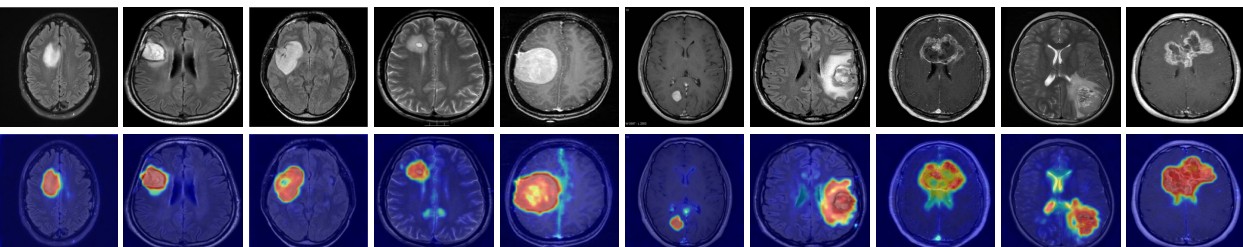

Figure 27: Localization score maps for the BrainMRI dataset. The first row illustrates the original image, while the second row shows the anomaly segmentation results, with the regions encircled in green representing the ground truth.

