# OpenReview forum: "Crane: Context-Guided Prompt Learning and Attention Refinement for Zero-Shot Anomaly Detection"
_TMLR — Rejected by TMLR_

### Review · Reviewer_SiE6 · 2026-03-03

**Summary Of Contributions:**

This paper proposes Crane./Crane+, a unified framework for zero-shot anomaly detection (ZSAD) that aims to address the persistent gap between image-level detection performance and pixel-level localization accuracy in CLIP-based methods.
Overall, the paper’s primary contribution lies in systematically addressing the limitations of CLIP’s global pretraining bias for anomaly localization, through coordinated improvements in attention refinement, prompt conditioning, and feature fusion. While individual components build upon established techniques, their integration into a unified anomaly-aware framework yields strong empirical gains in both detection and localization settings.

**Audience:**

Yes

**Audience Explanation:**

Anomaly detection has a wide range of real-world applications, and zero-shot anomaly detection in particular has even greater practical potential, since it does not require domain-specific labeled data. While many existing methods mainly focus on improving classification performance at the image level, providing accurate segmentation and localization of anomalous regions is often much more practical and useful in real-world scenarios.

In this regard, the paper’s emphasis on improving pixel-level localization makes the work more impactful and application-oriented. Therefore, I believe this work could attract interest not only from the TMLR community but also from a broader audience of researchers working on anomaly detection, domain generalization, and vision-language models.

**Claims And Evidence:**

Yes

**Claims Explanation:**

++ Strength

I think this paper has clear and convincing evidence for their contribution. The authors provide extensive experimental results across multiple datasets, including both industrial and medical domains, which support their main claims about improved image-level detection and pixel-level localization performance. The comparison with recent state-of-the-art methods is thorough, and the reported gains are consistent across different evaluation metrics.

--Weakness

While the method is well-designed, I have two main concerns. First, the framework introduces multiple modifications to both the vision and text encoders (attention refinement, prompt tuning, fusion, and DINO-based enhancement), which increases model complexity and may weaken the zero-shot generalization claim due to potential overfitting to auxiliary domains.

Second, the anomaly-aware local-to-global fusion relies on anomaly scores predicted by the same model, which may amplify errors or biases and raises questions about training stability and robustness.




Nevertheless, the extensive experimental results and the strong focus on improving segmentation performance make the contribution meaningful and practically relevant. In particular, emphasizing pixel-level localization addresses an important limitation of prior CLIP-based zero-shot anomaly detection methods.

As I do not have extensive experience reviewing or publishing in the specific area of anomaly detection with vision-language models, my evaluation is based on a more general perspective. I would appreciate it if this context is taken into consideration when interpreting my comments.

**Requested Changes:**

While the introduction motivates zero-shot anomaly detection in vision tasks, I felt that the broader scope of anomaly detection as a research field is not sufficiently contextualized. Anomaly detection is a very broad area that spans diverse domains beyond image analysis, such as time-series monitoring (e.g., current flow or sensor signals), network intrusion detection, financial fraud detection, and system reliability analysis.

Because of this wide applicability, it would help readers if the authors briefly clarified early on that this work specifically focuses on image-based anomaly detection, and more precisely on vision-language model-based zero-shot anomaly detection. Without this clarification, the scope of the problem may initially feel ambiguous.

Adding a short paragraph that positions the work within the broader anomaly detection landscape—while explicitly narrowing the focus to visual and zero-shot settings—would improve clarity and make the contribution easier to understand, especially for readers coming from other anomaly detection domains.

---

> ### Author Response · Authors · 2026-03-21
> **Author Response**
>
> We thank the reviewer for the constructive feedbacks and have incorporated all results and revisions into the paper and highlighted them in blue.
>
> **Pipeline Complexity and Zero-shot Generaliztion:** Thank you for raising this concern. Although our framework contains several components, they are designed to be complementary and lightweight rather than heavy task-specific adaptations. Moreover, several components, such as the score-based local-to-global fusion, are parameter-free, and the added modules are introduced to address different limitations of vanilla CLIP-based anomaly detection: context-guided prompt learning improves input-dependent alignment, extended self-correlation attention refines spatial features, and D-Attn enhances structurally meaningful localization. Importantly, our experiments across 14 datasets also support the zero-shot generalization claim, showing that the method generalizes consistently across diverse domains rather than relying on narrow dataset-specific fitting.
>
> **Training Stability and Robustness of Local-to-Global Fusion Module:** The anomaly-aware local-to-global fusion is introduced to filter and aggregate the most relevant patches from all local features when constructing the global representation. This is important because, in anomaly detection, the discriminative evidence is often confined to only a small subset of patches, while many other regions may be irrelevant or distracting. By using anomaly scores to guide this selection, the fusion step helps the model focus the global representation on anomaly-relevant local cues, leading to a more discriminative image-level feature. To address the reviewer’s concern about robustness, we additionally report performance across five different random seeds, which shows only a small standard deviation and supports the stability of our method.
>
> | Metric | Dataset | Ours (Crane) | Ours (Crane⁺) |
> |---|---|---|---|
> | Image-level ↑ (AUROC, AP, F1-max) | MVTec | (93.8 ± 0.36, 97.5 ± 0.13, 93.8 ± 0.32) | (93.9 ± 0.34, 97.6 ± 0.16, 93.6 ± 0.27) |
> | Image-level ↑ (AUROC, AP, F1-max) | VisA | (85.3 ± 0.28, 87.9 ± 0.24, 82.6 ± 0.25) | (83.6 ± 0.6, 86.7 ± 0.4, 81.2 ± 0.31) |
> | Pixel-level ↑ (AUROC, AUPRO, F1-max) | MVTec | (91.3 ± 0.19, 84.6 ± 0.84, 41.3 ± 0.64) | (91.2 ± 0.3, 88.1 ± 0.4, 43.8 ± 0.5) |
> | Pixel-level ↑ (AUROC, AUPRO, F1-max) | VisA | (95.1 ± 0.29, 87.5 ± 0.31, 30.9 ± 0.47) | (95.3 ± 0.11, 90.6 ± 0.32, 30.2 ± 0.084) |
>
> **Clarifying the Scope of the Work Early in the Introduction:** Thanks for the suggestion. We have changed our opening paragraph in the introduction as follows to clarify the scope of the paper right at the beginning:
>
> Image anomaly detection is the task where a set of training images representing normal visual patterns is given, and the goal is to identify test images that deviate from this notion of normality (Ruff et al. 2021). It is particularly important in practical scenarios where abnormal samples are rare, diverse, and difficult to collect during training.

---

### Review · Reviewer_4dX1 · 2026-03-08

**Summary Of Contributions:**

This paper proposes Crane, a CLIP-based framework for zero-shot anomaly detection. The method introduces a correlation-based attention module to enhance patch-level visual representations, a context-guided prompt learning strategy to adapt textual embeddings, and an anomaly-aware aggregation mechanism to emphasize anomalous regions. Experiments on multiple benchmarks demonstrate improvements.

**Audience:**

Yes

**Audience Explanation:**

The proposed E-Attn module addresses the limitation of CLIP patch representations for dense anomaly localization by introducing correlation-based attention across patches. The motivation is reasonable and the design is well aligned with the need for fine-grained spatial representations in anomaly detection.

**Broader Impact Concerns:**

No major ethical concerns.

**Claims And Evidence:**

No

**Claims Explanation:**

1. The motivation for context-guided prompt learning is unclear. Injecting the same image feature into both normal and abnormal text embeddings does not obviously improve their discriminability, and conditioning text representations on the input image weakens the role of text as external semantic knowledge in zero-shot settings.
2. The proposed anomaly-aware local-to-global fusion first computes patch–text similarity scores, uses these scores as weights to aggregate patch features, and then computes the similarity again. As a result, the final prediction effectively emphasizes patches with higher similarity scores, which can be interpreted as amplifying the anomaly scores rather than introducing a new feature fusion mechanism.
3. The ablation study in Table 2 (a,b,c,d) lacks a consistent baseline, making it difficult to clearly evaluate the contribution of each component. Different ablations appear to start from different model variants instead of incrementally adding modules to a unified baseline.

**Requested Changes:**

1. Please clarify the motivation and design of the context-guided prompt learning module. In particular, explain why injecting the same image feature into both normal and abnormal text embeddings improves discriminability, and discuss its impact on the zero-shot setting where text is typically expected to provide external semantic knowledge.
2. It would be helpful to clarify how this differs from simple score-based aggregation and whether it provides benefits beyond score amplification.
3. Please revise the ablation study to include a consistent baseline and incremental evaluation of each component. For example, start from a unified baseline and sequentially add the proposed modules to clearly demonstrate their individual and combined contributions.

---

> ### Author Response · Authors · 2026-03-21
> **Author Response**
>
> We thank the reviewer for the constructive feedbacks and have incorporated all results and revisions into the paper and highlighted them in orange.
>
> **Clarifying the Motivation Behind Context-Guided Prompt Learning, How It Helps, and Its Impact on Zero-Shot Generalization:** Rather than learning a single shared set of normal and abnormal prompts for all inputs, the model conditions the prompts on the visual context of each image. This produces text representations that are better aligned with the corresponding input features, enabling the model to learn normal and abnormal prompts in an instance-aware manner. For example, even within the same class, anomalies can appear in very different forms, such as a pill being broken, discolored, or scratched. Since many of these inputs still have nearly identical global features, providing the context through the image CLS token facilitates finer-grained alignment between the text and vision encoders, allowing the model to focus on the anomalous aspects that differ across instances. By making the prompts input-dependent, the method reduces reliance on dataset-specific prompt biases and improves generalization across visually diverse samples. We have clarified this in the ablation studies, highlighted in *red*, as this point was also shared by Reviewer 4Y7J.
>
> **Simple Anomaly Score Aggregation vs. Score-based Local-to-global Feature Fusion:** We thank the reviewer for suggesting this insightful ablation and further compare the introduced score-based feature aggregation with simple score aggregation. To do so, in Crane, we replace the score-based local-to-global feature fusion with score aggregation, in which instead of mean anomaly-aware feature, mean anomaly score of patches is calculated and averaged with image-level anomaly score. As shown in the table below, feature aggregation outperforms simple score aggregation on all image-level metrics. This suggests that averaging anomaly scores is less effective at capturing localized anomalies, since it is dominated by the most frequent patches in the image, which are usually normal. As a result, score aggregation is biased toward normality and reduces precision. Whereas, in feature aggregation the anomalous directions are retained and reflected in the final cosine similarity calculation with learned anomalous features. Moreover, local-to-global feature aggregation not only amplifies anomalous cues that are already captured in the image-level feature, but also incorporates anomalous regions ignored by the less anomaly sensitive attention of the original CLIP, providing richer and more holistic image-level feature.
>
> | Approach | MVTec-AD (AUROC, AP, F1-max) | VisA (AUROC, AP, F1-max) |
> |---|---|---|
> | Simple Score Aggregation | (92.7, 96.4, 93.2) | (84.1, 87.2, 81.8) |
> | Score-based Feature Aggregation | (93.8, 97.5, 93.8) | (85.3, 87.9, 82.6) |
>
> We have added this table to Appendix B.2 (Simple Anomaly Score Aggregation vs. Score-based Local-to-global Feature Fusion).
>
> **Clarifying the Different Impact of Score-based Aggregation and Context-Guided Prompt Learning:** We agree with the reviewer that our *Anomaly-aware Local-to-Global Fusion* increases anomaly sensitivity by emphasizing patches with higher anomaly relevance. The motivation is that a purely global CLIP image representation can be too coarse, since CLIP is trained to align images with relatively coarse text descriptions [Wysoczańska et al. (2024)]. In anomaly detection, normal and abnormal samples can therefore end up with very similar global representations. To address this, we fuse the global feature with local dense features that capture fine-grained abnormal regions. Since only a small subset of patches is typically informative, we first score the patches and then aggregate the most relevant ones. In this way, the global representation preserves overall semantic context, while the selected local features inject the critical fine-grained evidence.
>
> Context-guided prompt learning, however, plays a different role. Score-based aggregation operates on visual features: it selects and emphasizes the most relevant local patches before fusion. In contrast, context-guided prompt learning operates on the prompt side: it adapts the normal and abnormal prompts to the visual context of the current image. Its effect is therefore not just stronger anomaly emphasis, but better image-text alignment through input-dependent, instance-aware prompts. This allows the model to represent normality and abnormality more precisely for each sample, improving generalization beyond what score-based aggregation alone can provide.
>
> We thank the reviewer for the suggestion and have clarified this in the ablation studies, highlighted in *red*, as this point was also shared by Reviewer 4Y7J.

---

> ### Author Response · Authors · 2026-03-22
> **Author Response 2**
>
> **Providing Additional Incremental Ablation Study:** We have followed the standard ablation procedure, which removes one component each time from the pipeline and reports the performance to show the isolated impact of each contribution. However, to address the reviewers concern, we start from the baseline and report the performance each time a component is added. The table below shows the incremental ablation, starting from the AnomalyCLIP baseline and sequentially adding each proposed component. As shown, context-guided prompt learning improves both image- and pixel-level performance through better input-dependent alignment; extended self-correlation attention mainly improves pixel-level performance by refining spatial features; score-based aggregation primarily boosts image-level performance by enhancing the global representation while leaving dense predictions unchanged; and D-Attn provides the final refinement, especially for localization.
>
> | Module | MVTec AD Pixel-level (P-ROC, P-F1-max) | MVTec AD Image-level (I-ROC, I-F1-max) | VisA Pixel-level (P-ROC, P-F1-max) | VisA Image-level (I-ROC, I-F1-max) |
> |---|---|---|---|---|
> | Baseline (AnomalyCLIP) | (89.8, 36.9) | (91.0, 91.2) | (94.0, 25.2) | (80.1, 78.3) |
> | + Context-guided Prompt Learning | (90.6, 38.7) | (92.2, 92.3) | (94.5, 26.7) | (81.0, 79.2) |
> | + Extended Self-Correlation Attention | (91.2, 40.8) | (92.2, 92.3) | (94.3, 29.5) | (81.0, 79.2) |
> | + Score-based Aggregation | (91.2, 40.8) | (93.7, 93.9) | (94.3, 29.5) | (82.7, 80.4) |
> | + D-Attn | (92.1, 44.7) | (94.7, 94.3) | (95.5, 29.2) | (82.6, 80.6) |
>
> We have added this table to Appendix B.1 (Incremental Ablations) to provide a clearer understanding of the role of each proposed component.

---

### Review · Reviewer_4Y7J · 2026-03-11

**Summary Of Contributions:**

This paper studies zero-shot anomaly detection and localization, with a focus on improving pixel-level localization where prior CLIP-based methods remain weak. The proposed method, Crane, combines: (i) a correlation-based attention refinement for the CLIP vision encoder, (ii) context-guided prompt learning, and (iii) anomaly-aware local-to-global fusion. Crane+ further adds a DINOv2-guided spatial attention branch. Experiments on industrial and medical datasets show strong gains, especially on industrial pixel-level AUPRO.

The problem is important and the empirical results are promising. However, the paper is not yet fully convincing because the experimental protocol is not sufficiently clear, several design choices are only weakly justified, and the analysis remains shallow.

**Audience:**

Yes

**Audience Explanation:**

Yes. Zero-shot anomaly detection/localization is a relevant problem for the TMLR audience, and the paper addresses a practically important gap between image-level detection and pixel-level localization. The empirical findings, especially on localization, are likely to interest researchers working on multimodal models, dense prediction, and domain generalization.

**Broader Impact Concerns:**

I do not see a major unaddressed broader-impact issue.

**Claims And Evidence:**

No

**Claims Explanation:**

The paper provides extensive experiments, but the evidence does not fully support the strength of its claims.

First, the training/evaluation protocol is unclear. The paper states that the model is trained on MVTec-AD and evaluated on other datasets, but the appendix also says the method is fine-tuned on the test set of MVTec-AD, and trained on VisA test data when evaluating on MVTec-AD. This setting needs much clearer explanation, and it must be made explicit that all baselines follow exactly the same protocol.

Second, some claims are overstated. The paper suggests consistent state-of-the-art performance, but on medical pixel-level benchmarks the gains are mixed and seem strongest mainly on AUPRO rather than across all metrics.

Third, the paper mainly shows that the proposed components help, but does not explain well why they help. The design is intuitive, but the analysis does not yet provide enough insight into the underlying mechanism.

**Requested Changes:**

Critical: Clarify the exact experimental protocol, especially what data is used for training, validation, and testing for each benchmark.

Critical: Make clear that all baselines are evaluated under the same protocol.

Critical: Tone down claims of consistent state-of-the-art performance and reflect the mixed medical pixel-level results more accurately.

Critical: Better justify the main design choices and explain why they should improve localization.

Strengthening: Add deeper analysis of when the method helps and when it fails.

Strengthening: Report variance across multiple random seeds for the main results.

---

> ### Author Response · Authors · 2026-03-21
> **Author Response**
>
> We thank the reviewer for the constructive feedbacks and have incorporated all results and revisions into the paper and highlighted them in red.
>
> **Clarifying the Experimental and Evaluation Protocols:** We thank the reviewer for pointing out this inconsistency, which may have arisen from our use of the term “fine-tune” in Appendix A. We have replaced it with “train” (as there is no additional fine-tuning) and now explicitly state that, in this setting, the training data comes from the test split of the auxiliary dataset used for training.
> As described in Section 5.1 (Experiments Setting), we follow the training/evaluation protocol adopted in prior works (VAND, AnomalyCLIP, AdaCLIP, and AA-CLIP). Under this protocol, the model is trained on one dataset and evaluated on other non-overlapping target datasets. When evaluating on the same benchmark that serves as the main training dataset in this protocol, the model is instead trained on a different dataset with non-overlapping categories. Concretely, for VAND, AnomalyCLIP, AdaCLIP, and Crane, the main dataset is MVTec-AD and the auxiliary training dataset is VisA when evaluating on MVTec-AD itself. For AA-CLIP, we follow its default evaluation setting, in which VisA is the main dataset and MVTec-AD is the auxiliary training dataset, as this yields its best reported performance.
> We have revised the paper to make this protocol explicit for each benchmark and to clarify, for each baseline, the exact training and evaluation setting used for fair comparison.
>
> **Revising the tone of our claims and reflecting mixed medical pixel-level results:** We appreciate the reviewer’s insightful comment. In order to make the gains more precise, we have adjusted the tone, stating that the most consistent gains are achieved in the industrial settings. In the experiments section we highlighted the more mixed gains in pixel-level metrics of the medical domains, as well.
>
> **Elaborating further on the intuition behind the main design choices:** To address the reviewers’ concerns, we have further clarified the design motivations and revised the paper accordingly:
>
> *Anomaly-aware Local-to-Global Fusion:* A purely global image representation from the CLIP image encoder can be too coarse, as CLIP is trained to align images with relatively coarse text descriptions [Wysoczańska et al. (2024)]. In our setting, the normal and abnormal distributions can be very close, as illustrated in Figures 4, 5, and 8 of the appendix, leading to highly similar global representations. To address this, we fuse local dense features containing abnormal regions with the global representation, enabling the model to learn more discriminative prompts. Moreover, since the informative cues may be confined to only a few dense patches while large parts of the image are irrelevant or distracting, we first score the patches and then fuse only the most relevant ones. This motivates our combination of global and local information: the global representation preserves the overall semantic context of the image, while the local features allow the model to focus on fine-grained regions where the critical evidence appears. In this way, the two representations play complementary roles.
>
> *Context-guided Prompt Learning:* This contribution is inspired by [1]. Rather than learning a single shared set of normal and abnormal prompts for all inputs, the model conditions the prompts on the visual context of each image. This produces text representations that are better aligned with the corresponding input features, enabling the model to learn normal and abnormal prompts in an instance-aware manner. By making the prompts input-dependent, the method reduces reliance on dataset-specific prompt biases and improves generalization across visually diverse samples.
>
> *Extended Self-Correlation Attention + DINO-guided spatial Attention:* Our design is inspired by  CLIPSurgery Li et al. (2023) , which showed that standard CLIP self-attention often links semantically inconsistent regions, leading to noisy or misleading explanations, while query-query, key-key, and value-value attentions better capture relevant areas. Building on this, we replace the standard query-key attention with extended self-correlation modules that promote more semantically coherent token interactions, improving region localization. We further enhance this with DINO-guided self-spatial attention, which uses DINOv2 features to steer correlations toward spatially meaningful regions, especially for subtle anomalies. As also shown qualitatively in Figure 3 ( AnomalyCLIP vs. Crane vs. Crane+), this design suppresses distracting regions and better highlights abnormal areas.
>
> [1]- Conditional prompt learning for vision-language models. CVPR 2022.

---

> ### Author Response · Authors · 2026-03-21
> **Author Response 2**
>
> **Reporting the variance across multiple seeds:** Thanks for your suggestion. We report the standard deviation (std) across 5 seeds for 2 datasets and 6 measures, as shown below. We are running on other datasets and will include them in the final version as well:
>
> | Metric | Dataset | Ours (Crane) | Ours (Crane⁺) |
> |---|---|---|---|
> | Image-level ↑ (AUROC, AP, F1-max) | MVTec | (93.8 ± 0.36, 97.5 ± 0.13, 93.8 ± 0.32) | (93.9 ± 0.34, 97.6 ± 0.16, 93.6 ± 0.27) |
> | Image-level ↑ (AUROC, AP, F1-max) | VisA | (85.3 ± 0.28, 87.9 ± 0.24, 82.6 ± 0.25) | (83.6 ± 0.6, 86.7 ± 0.4, 81.2 ± 0.31) |
> | Pixel-level ↑ (AUROC, AUPRO, F1-max) | MVTec | (91.3 ± 0.19, 84.6 ± 0.84, 41.3 ± 0.64) | (91.2 ± 0.3, 88.1 ± 0.4, 43.8 ± 0.5) |
> | Pixel-level ↑ (AUROC, AUPRO, F1-max) | VisA | (95.1 ± 0.29, 87.5 ± 0.31, 30.9 ± 0.47) | (95.3 ± 0.11, 90.6 ± 0.32, 30.2 ± 0.084) |
>
> **Add deeper analysis of when the method helps and when it fails:**  Thanks for the suggestion. We have included an extensive qualitative comparison with state-of-the-art methods in Figures 4–27 of the appendix, showing that our method is particularly effective at detecting small anomalies compared to prior methods. In addition, Figure 3 qualitatively compares AnomalyCLIP, our starting baseline, Crane, and Crane+ across a diverse set of inputs with different anomaly types (texture and object) and varying anomaly sizes. As shown, Crane produces anomaly maps with less noise and tighter boundaries, indicating fewer false positives and higher localization accuracy. We attribute these improvements to our extended self-correlation attention module, anomaly-aware local-to-global fusion, and context-guided prompt learning. Crane+ further improves the anomaly maps by leveraging DINOv2 knowledge to better filter false-positive regions. Regarding failure cases, we have already discussed several limitations in Appendix G.2. In addition, as shown in Table 1, incorporating DINOv2 can in some cases slightly reduce image-level performance, for example on MPDD and VisA. We attribute this to the misalignment between the frozen DINOv2 features and the CLIP vision encoder. Addressing this, for example by unfreezing this module, is an interesting direction that we leave for future work. We are also conducting additional experiments to better highlight both the strengths and failure cases of the method, and we plan to include them in the camera-ready version if they provide additional insight.

---

### Author Response · Authors · 2026-03-21
**Message to All Reviewers**

We sincerely thank all reviewers for recognizing the merits of our work and for providing constructive feedback. Thanks to your comments, the paper has been significantly improved. In the revised version, we have incorporated the requested changes and highlighted them using color coding for each reviewer. We will remove the color coding in the camera-ready version. We hope we have adequately addressed all concerns, and we would be grateful for any further comments if anything remains unclear.

---

### Author Response · Authors · 2026-03-23
**Message to All Reviewers**

Dear Reviewers,

We have dedicated considerable effort to conducting further experiments and meticulously addressing each of your comments. We are eager to hear your thoughts on our rebuttal and fully prepared to engage in further discussions regarding any additional concerns you may have.

Warm regards, The Authors.

---

### Author Response · Authors · 2026-04-04
**Message to All Reviewers**

Dear Reviewers,

Thank you again for your time and effort. It has been about a week since we posted our rebuttal, and we would greatly appreciate any feedback you may have. As the deadline for the final decision is approaching, we kindly ask you to share your comments when convenient.

Warm regards,
The Authors

---

### Decision · Action_Editor_TmwA · 2026-05-08

**Recommendation:** Reject

**Audience:**

Yes

**Audience Explanation:**

Zero-shot anomaly detection and localization is a relevant problem for the TMLR audience, particularly for researchers working on vision-language models, dense prediction, anomaly detection, and domain generalization.

**Claims And Evidence:**

No

**Claims Explanation:**

After the rebuttal, the AE and reviewers believe that several core claims are not sufficiently supported. The motivation for context-guided prompt learning remains unclear, as injecting the same image feature into both the normal and abnormal text embeddings does not obviously improve class discriminability and may instead introduce a shared bias. The anomaly-aware local-to-global fusion appears closer to score reweighting, and the comparison with uniform averaging is insufficient to justify it as a standalone contribution. Although the authors clarified several points and added additional ablations, the analysis mostly shows that each component improves performance without convincingly explaining why.